# Nuclear PYHIN proteins target the host transcription factor Sp1 thereby restricting HIV-1 in human macrophages and CD4+ T cells

**Matteo Bosso[1], Caterina Prelli Bozzo[1], Dominik Hotter[1], Meta Volcic[1], Christina M. Stürzel[1], Annika Rammelt[1], Yi Ni[2], Stephan Urban[2], Miriam Becker[3], Mario Schelhaas[3], Sabine Wittmann[4], Maria H. Christensen[5], Florian I. Schmidt[5], Thomas Gramberg[4], Konstantin M. J. Sparrer[1], Daniel Sauter[1], Frank Kirchhoff[1] ***

**1** Institute of Molecular Virology, Ulm University Medical Center, Ulm, Germany, **2** Department of Infectious Diseases, Molecular Virology, University Hospital Heidelberg, Heidelberg, Germany, **3** Institute of Cellular Virology, ZMBE, University of Münster, Münster, Germany, **4** Institute of Clinical and Molecular Virology, Friedrich-Alexander University Erlangen-Nürnberg, Erlangen, Germany, **5** Institute of Innate Immunity, University of Bonn, Bonn, Germany

* frank.kirchhoff@uni-ulm.de

**Data Availability Statement:** All relevant data are within the manuscript and its Supporting Information files.

## Abstract

Members of the family of pyrin and HIN domain containing (PYHIN) proteins play an emerging role in innate immunity. While absent in melanoma 2 (AIM2) acts a cytosolic sensor of non-self DNA and plays a key role in inflammasome assembly, the γ-interferon-inducible protein 16 (IFI16) restricts retroviral gene expression by sequestering the transcription factor Sp1. Here, we show that the remaining two human PYHIN proteins, i.e. myeloid cell nuclear differentiation antigen (MNDA) and pyrin and HIN domain family member 1 (PYHIN1 or IFIX) share this antiretroviral function of IFI16. On average, knock-down of each of these three nuclear PYHIN proteins increased infectious HIV-1 yield from human macrophages by more than an order of magnitude. Similarly, knock-down of IFI16 strongly increased virus transcription and production in primary CD4+ T cells. The N-terminal pyrin domain (PYD) plus linker region containing a nuclear localization signal (NLS) were generally required and sufficient for Sp1 sequestration and anti-HIV-1 activity of IFI16, MNDA and PYHIN1. Replacement of the linker region of AIM2 by the NLS-containing linker of IFI16 resulted in a predominantly nuclear localization and conferred direct antiviral activity to AIM2 while attenuating its ability to form inflammasomes. The reverse change caused nuclear-to-cytoplasmic relocalization of IFI16 and impaired its antiretroviral activity but did not result in inflammasome assembly. We further show that the Zn-finger domain of Sp1 is critical for the interaction with IFI16 supporting that pyrin domains compete with DNA for Sp1 binding. Finally, we found that human PYHIN proteins also inhibit Hepatitis B virus and simian vacuolating virus 40 as well as the LINE-1 retrotransposon. Altogether, our data show that IFI16, PYHIN1 and MNDA restrict HIV-1 and other viral pathogens by interfering with Sp1-dependent gene expression and support an important role of nuclear PYHIN proteins in innate antiviral immunity.

**Funding:** This work was supported by grants from the German Research Foundation (DFG CRC 1279 and SPP 1923); F.I.S is funded through the Emmy Noether Programme (SCHM 3336-1-1) and Germany's Excellence Strategy – EXC2151 – 390873048. This investigation used resources supported by the Southwest National Primate Research Center grant P51 OD011133 from the Office of Research Infrastructure Programs, National Institutes of Health. The funders had no role in study design, data collection and analysis, decision to publish, or preparation of the manuscript.

**Competing interests:** NO authors have competing interests.

## Author summary

Pyrin and HIN domain (PYHIN) proteins are best known for their role in immune sensing and inflammasome activation. However, most studies focused on the cytosolic DNA sensor AIM2. The human genome encodes four different PYHIN proteins, and recent studies revealed that the nuclear PYHIN protein IFI16 suppresses transcription of HIV and other viral pathogens. Here, we show that two other members of this protein family, MNDA and PYHIN1, that are also predominantly found in the nucleus share the antiretroviral activity of IFI16. Importantly, our data demonstrate that nuclear PYHIN proteins restrict HIV-1 in primary human macrophages and CD4+ T lymphocytes. Functional analyses revealed that the pyrin domains of these factors compete with DNA for binding of the transcription factor Sp1. Altogether, our results support a key role of nuclear PYHIN proteins in restricting HIV-1 and suggest that they are also active against other viral pathogens and retrotransposons.

## Introduction

Pyrin and HIN domain (PYHIN) proteins are interferon(IFN)-inducible factors playing key roles in innate immunity [1,2]. One characteristic feature is an N-terminal pyrin domain (PYD) involved in interactions with other PYD-containing proteins, forming complexes with roles in apoptosis and inflammasome activation [3,4]. In addition, PYHIN proteins share one or several HIN-200 domains allowing double-stranded DNA binding [5]. Finally, some cellular factors contain just a single PYD, and these Pyrin-only proteins (POPs) were reported to suppress inflammasome formation in response to microbial infections [6,7]. The number of PYHIN proteins varies between mammalian species. Humans encode four PYHIN proteins: absent in melanoma 2 (AIM2), IFN-γ inducible protein 16 (IFI16), myeloid cell nuclear differentiation antigen (MNDA), pyrin and HIN domain family member 1 (PYHIN1/IFIX). IFI16 contains a bipartite nuclear localization signal (NLS) adjacent to the PYD and is therefore mainly located within the nucleus [8]. Similarly, PYHIN1 and MNDA show a predominant nuclear localization mediated by at least one putative NLS [9,10]. In contrast, AIM2 lacks an NLS and is predominantly found in the cytoplasm.

Human PYHIN proteins are best established as immune sensors of foreign DNAs [1,11]. Specifically, AIM2 is known to interact with microbial DNA to form caspase-1-activating inflammasomes [12], while IFI16 may boost immune sensing and promote type I IFN gene expression [13,14]. Compared to AIM2 and IFI16, MNDA and PYHIN1 are poorly investigated. MNDA is only expressed in myeloid-derived cells [15] and thought to bind DNA [16] via its HIN domain [17]. MNDA is upregulated by IFN-α in monocytes and interacts with proteins involved in transcriptional regulation of macrophage differentiation and activation [15,18]. PYHIN1 is expressed in several isoforms in epithelial cell lines [9], as well as in CD4+ lymphocytes [19] and primary foreskin fibroblasts [20]. PYHIN1 binds DNA in a sequence–independent manner, triggers IFN-β transcription when over-expressed in HEK293 cells [10] and acts as a tumor suppressor gene by increasing p53 stability and by upregulating the metastasis-suppressor Maspin [9,21,22].

Accumulating evidence suggests that PYHIN proteins not only sense microbial DNAs to induce IFNs and/or inflammasome activation [1,2] but also inhibit viral pathogens more directly. Especially, IFI16 is emerging as important antiviral restriction factor. IFI16 is mainly found in the nucleus and has been reported to interact with nuclear herpesviral DNAs [23–25] and to inhibit transcription of human cytomegalovirus (HCMV) [26], herpes simplex virus 1

(HSV-1) [27,28] and human papilloma virus (HPV) [29]. Recently, also PYHIN1 was shown to inhibit HSV-1 replication by reducing viral gene expression [20], although the underlying mechanism remains elusive. Following up on these findings and our observation that IFI16 shares properties of known antiretroviral restriction factors [30], we recently showed that IFI16 suppresses gene expression of HIV-1 and other primate lentiviruses as well as LINE-1 retroelements by interfering with the availability of the transcription factor Sp1 [31]. Here, we show that the antiviral activity and mechanism of IFI16 are shared by the two predominantly nuclear human PYHIN proteins IFIX/PYHIN1 and MNDA, but not the cytoplasmic AIM2. We also demonstrate that nuclear PYHIN proteins restrict primary HIV-1 strains in CD4+ T cells and macrophages representing the major viral target cells *in vivo*.

## Results

### Nuclear human PYHIN proteins inhibit HIV-1 transcription

We have previously shown that IFI16 inhibits HIV-1 transcription [31]. Two other human PYHIN proteins (PYHIN1/IFIX and MNDA) also contain the PYD and nuclear localization signal (NLS) that are critical and sufficient for antiretroviral activity of IFI16 (Fig 1A). For functional analyses, we generated constructs coexpressing the blue fluorescent protein (BFP) and HA-tagged versions of IFI16, PYHIN1 and MNDA, as well as AIM2, which naturally lacks an NLS. Western blot analysis confirmed that all four human PYHIN proteins are expressed at detectable levels in transfected HEK293T cells (Fig 1B). Real-time quantitative PCR revealed that PYHIN1 and MNDA suppress the production of initial (R-U5/*gag* containing mRNA) as well as nearly (*nef*) and fully (U3-polyA) completed HIV-1 RNA transcripts in transfected HEK293T cells as efficiently as IFI16, while AIM2 displayed little or no inhibitory activity (Fig 1C). In agreement with the effect on viral transcripts, expression of IFI16, PYHIN1 and MNDA inhibited infectious virus yield in a dose-dependent manner (Fig 1D, left). Reduction of infectious virus yield was associated with inhibition of LTR-driven eGFP expression from HIV-1 NL4-3-IRES-eGFP proviral constructs (Fig 1D, right). As previously observed for IFI16 [31], the inhibitory effect of PYHIN1 was more pronounced if LTR-activity was boosted by the viral transactivator Tat. None of the human PYHIN proteins had a significant inhibitory effect on the activity of the HCMV major immediate early promoter (MIEP) (Fig 1E). Additional control experiments showed that the HCMV MIEP activity is enhanced by increasing levels of p65/NF-κB, but not Sp1 (S1 Fig). In contrast, overexpression of Sp1 increased HIV-1 NL4-3 LTR promoter activity up to 10-fold (S1A Fig).

IFI16 has been reported to act as immune sensor for viral DNAs triggering IFN production by a cGAS- and STING-dependent mechanism [13,14,23,32]. Experiments in HEK293T cells naturally lacking endogenous STING expression revealed that IFI16 inhibits HIV-1 independently of interferon induction [31,33]. Consistent with this, expression of all four human PYHIN proteins alone or together with HIV-1 in HEK293T cells did not result in significant type I IFN induction (Fig 2). In contrast, IFN expression was induced when STING was transiently expressed or after infection with Sendai virus (SeV) activating innate immune responses in a STING-independent manner. Notably, IFI16, PYHIN1 and MNDA all significantly boosted IFN production in the presence of STING, while AIM2 had no enhancing effect (Fig 2). Thus, similarly to IFI16 [14,31], PYHIN-1 and MNDA restrict HIV-1 independently of IFN induction by suppressing proviral transcription but may also promote STING-dependent IFN responses.

### Human PYHIN proteins restrict HIV-1 in primary macrophages

IFI16 is expressed in primary HIV-1 target cells and its knock-down in macrophages increased infectious virus production [31]. To assess the possible relevance of other human PYHIN

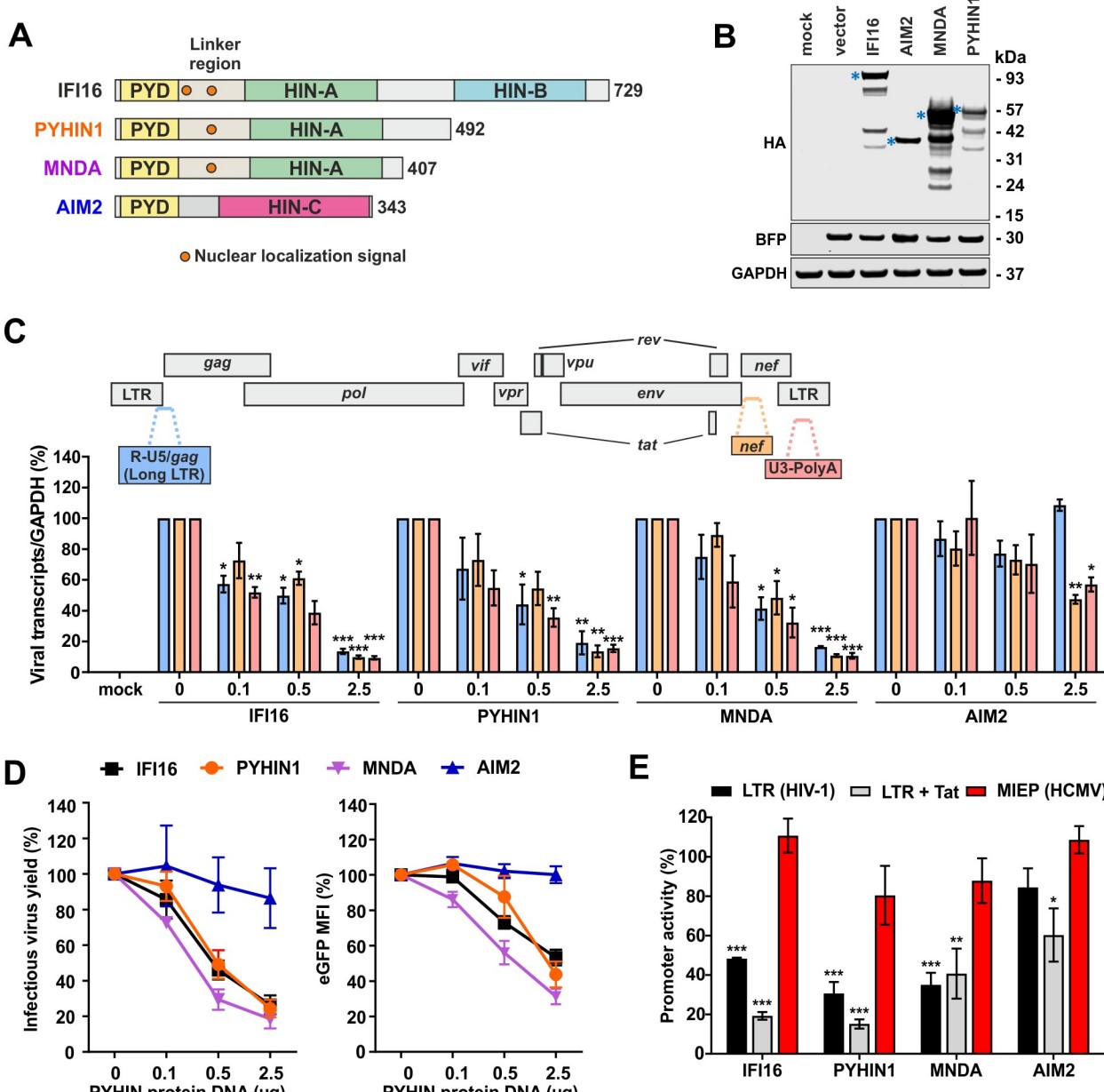

**Fig 1. Nuclear human PYHIN proteins inhibit HIV-1 transcription.** (A) Schematic presentation of the domain structure of human PYHIN proteins. Numbers indicate amino acid positions. (B) Expression of human PYHIN proteins. HEK293T cells were transfected with IRES-GFP constructs expressing the indicated HA-tagged PYHIN proteins. GFP and GAPDH were used as controls. Stars indicates bands of the expected size. (C) Levels of indicated viral transcripts in HEK293T cells cotransfected with increasing doses of expression constructs for IFI16, MNDA, PYHIN1 and AIM2 and constant quantities of the proviral HIV-1 NL4-3 construct determined by qRT-PCR in multiplex reactions. Data show mean percentages (±SEM; n = 3) relative to those detected in the presence in the empty control plasmid (100%). * p <0.05, ** p <0.01, *** p <0.001. Primers and probes were designed as previously described [65]. (D) Infectious virus yields in cell supernatants (left) and levels of LTR-dependent eGFP expression (right) in HEK293T cells transfected with vectors coexpressing the indicated PYHIN proteins and eGFP and the proviral HIV-1 NL4-3 IRES-eGFP construct were determined by TZM-bl cell infection assay and flow cytometric analysis at two days post-transfection. (E) HEK293T cells were cotransfected with luciferase reporter constructs under the control of the indicated promoters (MIEP, major immediate early promoter) and expression constructs for IFI16, PYHIN1, MNDA and AIM2 or a vector control. A construct expressing NL4-3 Tat under the control of the CMV IE promoter was used to activate the LTR. Shown are mean values (±SEM) measured in the presence of PYHIN proteins relative to the vector control (100%) obtained in three experiments.

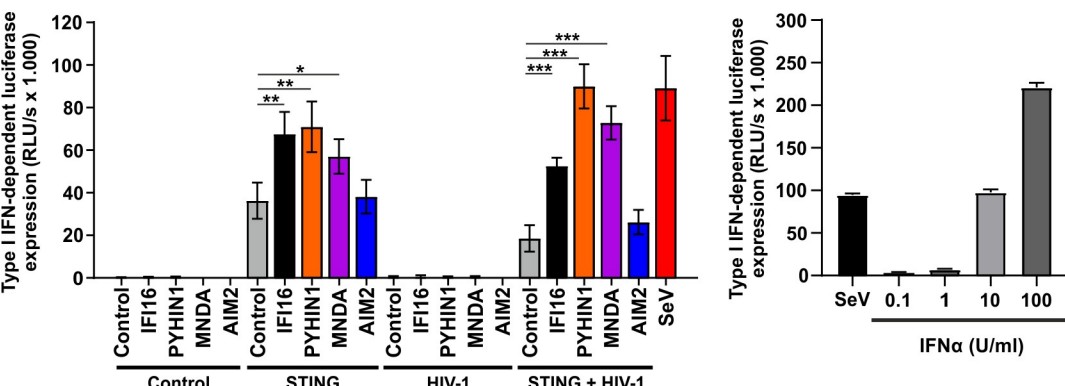

**Fig 2. Human PYHIN proteins do not induce IFN in HEK293T cells lacking STING.** HEK293T cells were transfected with the indicated PYHIN protein and/or STING expression vectors and the proviral HIV-1 NL4-3 IRES-eGFP construct, or infected with Sendai virus (SeV). 40 hours later, supernatants were harvested and used to stimulate HL116 IFN-reporter cells in triplicates. After 8 hours, luciferase expression in the HL116 cells was determined. Shown are mean values from three independent experiments ± SD. P-values indicate differences between IFN production detected in the presence and absence of the indicated PYHIN proteins; * p <0.05, ** p <0.01, *** p <0.001. The right panel shows an IFNα standard curve for comparison.

proteins in HIV-1 replication, we examined PYHIN1, MNDA and AIM2 expression in human CD4+ T cells and monocyte-derived macrophages (MDM) treated with various stimuli. In agreement with our previous study [31], three isoforms of IFI16 were detected and moderately induced by type I IFNs in CD4+ T cells and MDM (Figs 3A and S2A). Three isoforms of PYHIN1 (α, β and γ) were readily detectable in MDM but none of them was further upregulated by IFNs. In comparison, the α isoform was absent, and only low levels of the β and γ forms of PYHIN1 were detectable in unstimulated CD4+ T cells. However, the β form of PYHIN1 was induced by stimulation with anti-CD3/CD28 beads mimicking stimulation by antigen-presenting cells but not by IL-2 or IFNs (Figs 3A and S2A). As expected from published data [15], MNDA was detected in macrophages but not in T cells and significantly induced by IFN- α but not by IFN-γ (Figs 3A and S2A). AIM2 is expressed in both T cells and (at higher levels) MDM but was not further inducible by IFN treatment.

Our expression analyses suggested that IFI16 and (to a lesser extent) PYHIN1 may restrict HIV-1 in activated CD4+ T cells, while all three nuclear PYHIN proteins may cooperate in inhibiting retroviral replication in macrophages. To examine the latter, we performed IFI16, MNDA and PYHIN1 siRNA knock-down (kd) studies in macrophages from nine human donors (example shown in S2B Fig). The knock-down efficiencies varied between different donors and were on average more effective for MNDA (~80% reduction) than for IFI16, PYHIN1 and AIM2 (all ~50%) (Fig 3B). We also examined whether HIV-1 infection affected expression of these PYHIN proteins. Unexpectedly, the levels of MNDA expression were significantly reduced in virally infected cultures, while the expression levels of IFI16, PYHIN1 and AIM2 did not differ significantly between HIV-1-infected and uninfected macrophage cultures (Fig 3C). Despite moderate kd efficiencies, reduced IFI16 and PYHIN1 expression increased infectious virus yield of CCR5-tropic HIV-1 AD8, which is capable of spreading infection in macrophages, on average by ~6.4-fold at 3 dpi and ~21.6-fold at 6 dpi (Fig 3D). In comparison, kd of MNDA resulted in 3.5- and 16.1-fold higher infectious virus yields, while kd of AIM2 had no significant enhancing effect (Fig 3D). Combined knock-down of all three nuclear PYHIN proteins achieved 7.2- and 26.7-fold higher infectious virus yields at 3- and 6-days post-infection, respectively. These data show that IFI16, PYHIN1 and MNDA significantly restrict HIV-1 in primary human macrophages.

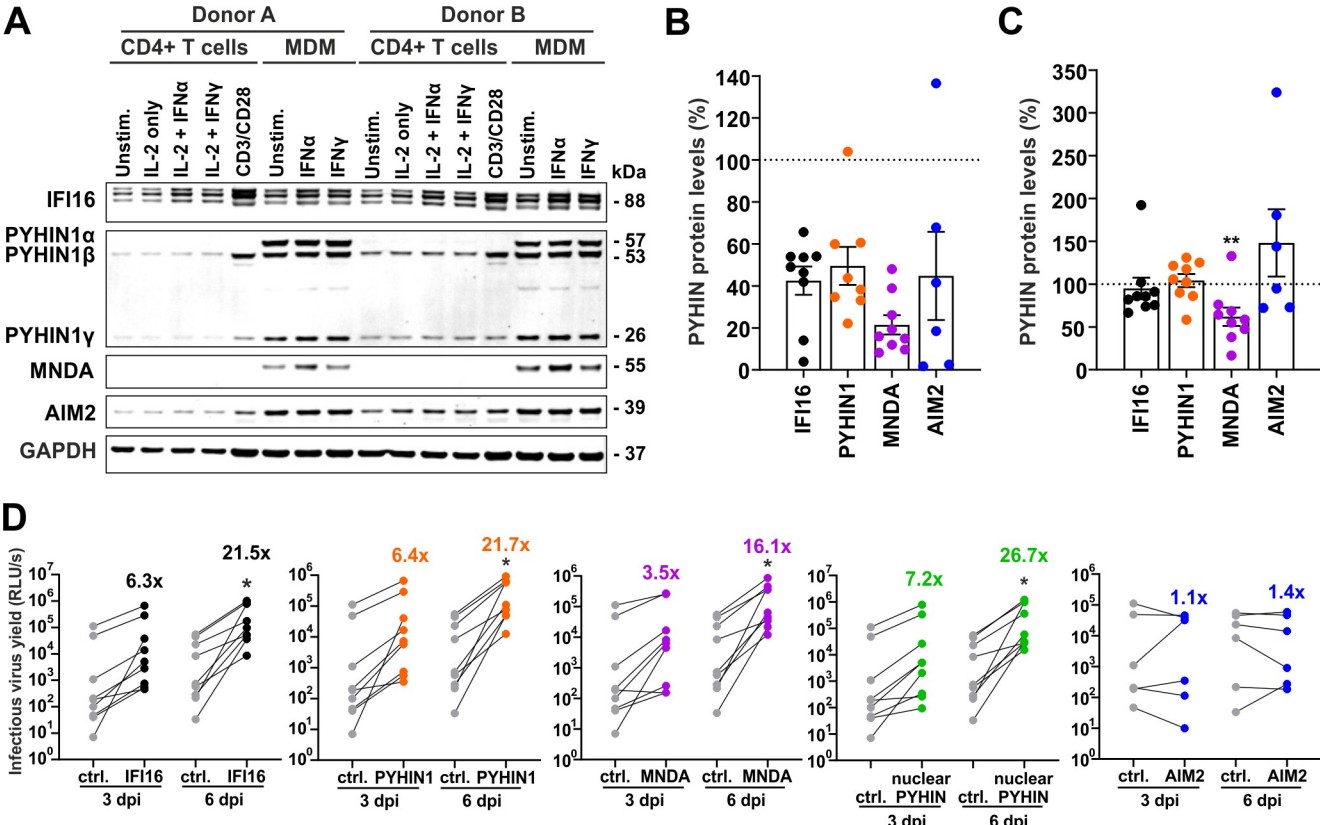

**Fig 3. Nuclear human PYHIN proteins restrict HIV-1 in human macrophages.** (A) Expression of PYHIN proteins in primary HIV-1 target cells. CD4+ T lymphocytes were stimulated with IL-2 alone, IL-2 + IFNα (500 U/ml), IL-2 + IFNγ (200 U/ml) or IL-2 + anti-CD3/CD28 beads. Macrophages (MDM) were stimulated with either IFNα (500 U/ml) or IFNγ (200 U/ml). 72 hours post stimulation, cells were harvested and lysed for Western Blot analysis. (B). Knockdown efficiencies in human MDM. Cells were treated with PYHIN-specific or control siRNA. Each point represents protein expression levels compared to cells treated with control siRNA (100%) obtained from one donor. (C) Expression levels of the indicated PYHIN proteins in HIV-1 infected compared to uninfected MDM cultures (100%). **, p<0.01. (D) Infectious virus production at 3 or 6 days post-infection (dpi) from MDM treated with the indicated siRNAs. In the nuclear PYHIN panel, siRNAs targeting IFI16, PYHIN1 and MNDA were combined. Numbers give average n-fold enhancement of infectious virus yield compared to cells treated with the control siRNA.

## IFI16 restricts HIV-1 and interacts with Sp1 in primary CD4+ T cells

CD4+ T cells do not express MNDA, and the levels of PYHIN1 are lower than in macrophages, while IFI16 is efficiently expressed in both cell types (Fig 3A). In our previous study [31], we did not examine whether IFI16 restricts HIV-1 in CD4+ T cells since no methods for effective knock-down in this cell type were available. More recently, we accomplished this by two different approaches. Nucleofection of CRISPR-Cas9 complexes containing IFI16 specific guide RNAs into activated CD4+ T cells [34] reduced the levels of IFI16 expression by ~70% without affecting PYHIN1 expression levels (Fig 4A). Strikingly, partial knock-out (KO) of IFI16 in primary CD4+ T cells increased infectious yield of the HIV-1 subtype B CH040 and CH058 transmitted-founder (TF) strains up to 20-fold but had no significant effect on the HIV-1 subtype C CH042 IMC (Fig 4B). This result agrees with our previous finding that subtype C HIV-1 strains are relatively resistant to IFI16 because they are less dependent on Sp1 for efficient transcription [31]. Measurement of p24 capsid antigen levels in the culture supernatants confirmed that reduced IFI16 expression increased CH040 production up to 20-fold but had no significant effect on CH042 (Fig 4C). Real-time quantitative PCR showed that IFI16 knock-down increased the production of initial as well as near and fully completed HIV-1 RNA

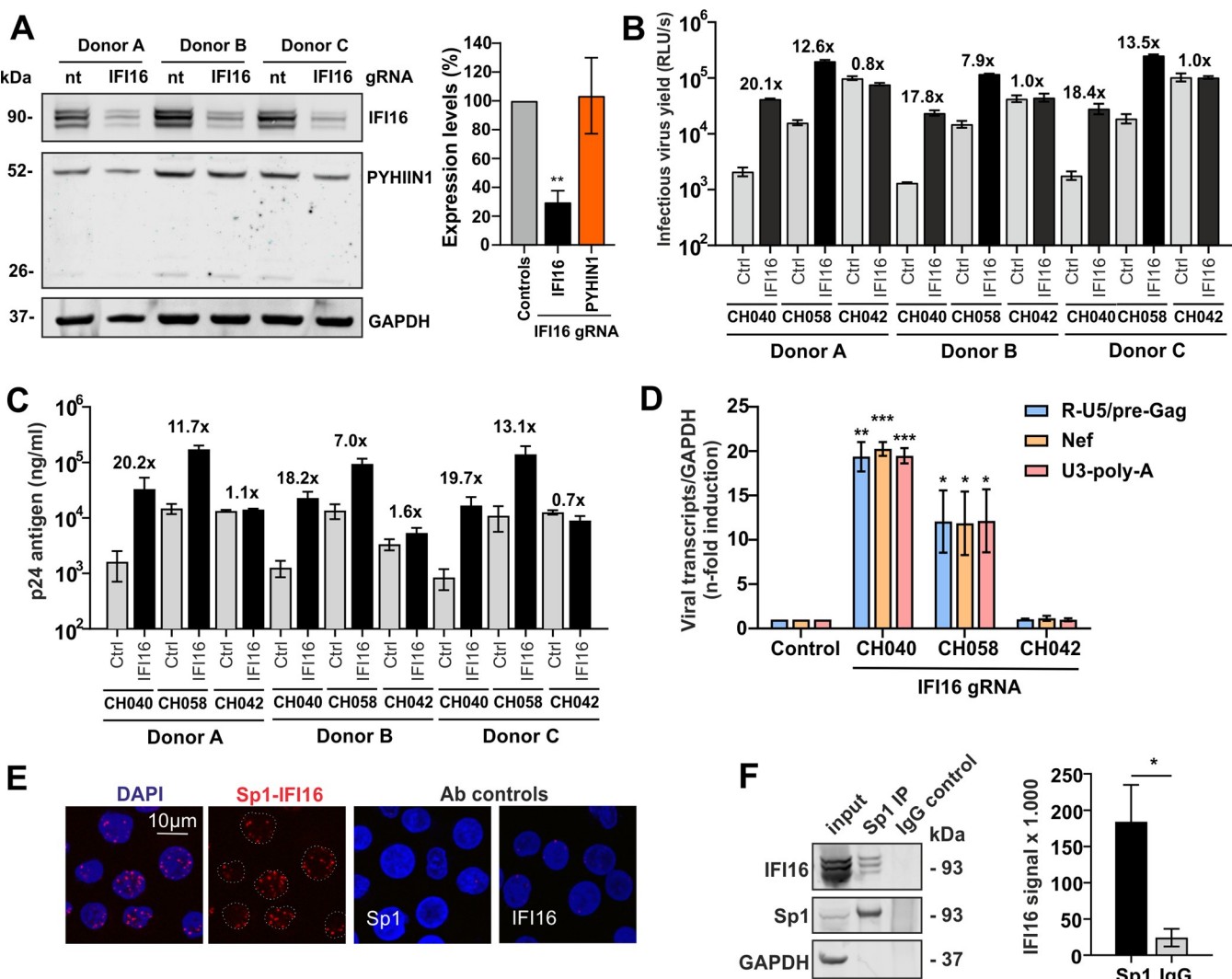

**Fig 4. IFI16 restricts HIV-1 and interacts with Sp1 in primary CD4+ T lymphocytes.** Endogenous IFI16 inhibits HIV-1 in primary CD4+ T lymphocytes. Cells from three healthy donors were isolated and activated with IL-2 and anti-CD3/CD28 beads. 72 hours post-activation, cells were transfected with Cas9 in complex with either a non-targeting (nt) or an IFI16-specific gRNA. At 96 hours post-transfection, cells were transduced with the indicated VSV-G pseudotyped HIV-1 strains. (A-D) The reduction of IFI16 protein levels in infected cell cultures (A), the infectious virus yield (B), p24 antigen production (C) and levels of viral RNA transcripts (D) were determined by Western blot, TZM-bl infection assay, ELISA and qRT-PCR, respectively, three days post infection. Bar diagrams in panel A and D show mean values (±SD) of the three different donors; those in panel B and C from triplicate measurements. Numbers above bars indicate n-fold change between cells treated with control or IFI16 specific gRNA. * p <0.05, ** p <0.01, *** p <0.001. (E) Association of Sp1 and IFI16 by *in situ* PLA. (F) Sp1 co-precipitates with IFI16 and PYHIN in CD4+ T lymphocytes. Cells from three healthy donors were isolated and activated with IL-2 and anti-CD3/CD28 beads. 72 hours post activation, cells were lysed and endogenous Sp1 was immunoprecipitated using magnetic beads coated with either an Sp1 antibody or control IgG. Co-IP eluates and input controls were subsequently analysed by Western Blotting. Shown is the blot of one representative experiment. On the right-hand panel, the IFI16 signal intensity from three independent experiment (±SEM) is shown.

transcripts in primary CD4+ T cells ~20-fold in the case of HIV-1 CH040 and ~12-fold for CH058 but had no enhancing effect on CH042 (Fig 4D). Similar results on infectious virus yield (S3A Fig), p24 antigen production (S3B Fig) and viral RNA levels (S3C Fig) were obtained when we used an shRNA-based approach. However, the magnitude of the effects on infectious HIV-1 yield was lower because IFI16 expression levels were only reduced by 30% to 50% (S3D Fig). Our previous studies in macrophages and in transfected HEK293T cells showed that HIV-1 subtype C strains like CH042 are largely resistant to IFI16 because they are less dependent on Sp1 for effective transcription than other subtypes of HIV-1 [31] and the

present results in CD4+ T cells agree with this. To further examine whether IFI16 targets Sp1 to inhibit HIV-1, we performed proximity ligation assays (PLA) for endogenous IFI16 and Sp1 in primary CD4+ T cells. We detected a significant number of PLA signals (Fig 4E) showing that IFI16 and Sp1 are in close proximity in the nucleus. In support of an interaction, Sp1 co-immunoprecipitated endogenous IFI16 (Fig 4F). Altogether, these data demonstrate that endogenous IFI16 efficiently restricts primary HIV-1 subtype B strains and interacts with Sp1 in human CD4+ T cells.

## The PYD and NLS of human PYHIN proteins are sufficient for HIV restriction

One surprising finding of our previous study was that the N-terminal PYD and NLS-containing linker region are sufficient for anti-HIV-1 activity of IFI16, whereas the two HIN domains involved in viral DNA interaction are dispensable [31]. To examine whether the same domains are critical for antiretroviral activity of other human PYHIN proteins, we generated constructs expressing HA-tagged forms of the PYD-only and PYD plus linker region of PYHIN1, MNDA and AIM2 (Fig 5A). In agreement with the findings on IFI16, the N-terminal PYD plus linker region of MNDA and PYHIN1 displayed significant activity against HIV-1 (Figs 5B and S4A) without inducing cytotoxic effects (S4B Fig). In the case of PYHIN1, the PYD plus linker region mutant was even more active than the full-length protein. The effect of the parental and mutant IFI16, AIM2, PYHIN1 and MNDA proteins on infectious virus yield correlated with their impact on LTR-driven eGFP expression from the proviral HIV-1 constructs ($R^2 = 0.914$; $p<0.0001$), further supporting that suppression of transcription plays a key role in reduced virus production.

## The NLS of IFI16 confers anti-HIV activity to AIM2

It has been shown that nuclear localization is critical for the antiretroviral activity of IFI16 [31]. Lack of the NLS containing linker reduced the antiretroviral activity of IFI16, MNDA and PYHIN1 (Fig 5B). However, the N-terminal PYD of all human PYHIN proteins inhibited HIV-1 to some extent. Even the PYD of AIM2 that is highly divergent from those of IFI16, MNDA and PYHIN1 [35] (S4C Fig) displayed antiviral activity although the full-length form was inactive. Thus, we hypothesized that the antiretroviral activity of the PYD might be conserved between human PYHIN proteins and that this domain alone might be small enough to passively enter into the nucleus in the absence of an NLS. To examine whether translocation of AIM2 to the nucleus confers antiretroviral activity, we swapped the linker region of IFI16 that contains NLS signals with that of AIM2 lacking an NLS (Fig 6A). All IFI16 and AIM2 variants were expressed at detectable levels (Fig 6B). The linker region of IFI16 rendered AIM2 active against HIV-1, while the reverse change disrupted this function of IFI16 (Fig 6C). For reasons that remain to be determined, the IFI16 construct containing the AIM2 linker moderately increased LTR-dependent GFP expression. Confocal microscopy analyses (Fig 6D) confirmed that the linker region of AIM2 localized IFI16 to the cytoplasm, while the NLS of IFI16 resulted in a predominantly nuclear localization of AIM2 (Fig 6E). These data further underline the importance of nuclear localization for antiretroviral activity of human PYHIN proteins. In addition, the results show that despite limited amino acid sequence homology (S4C Fig), the antiretroviral activity of the PYD is conserved between the nuclear PYHIN proteins IFI16, PYHIN1 and MNDA as well as the cytoplasmic immune sensor AIM2.

## IFI16 does not nucleate inflammasomes

In distinct sentinel cells of the innate immune system, cytosolic DNA can trigger the assembly of AIM2 inflammasomes: AIM2 oligomerizes on exposed DNA and nucleates the polymerization and cross-linking of the adaptor protein ASC [36–38]. This results in the assembly of ASC

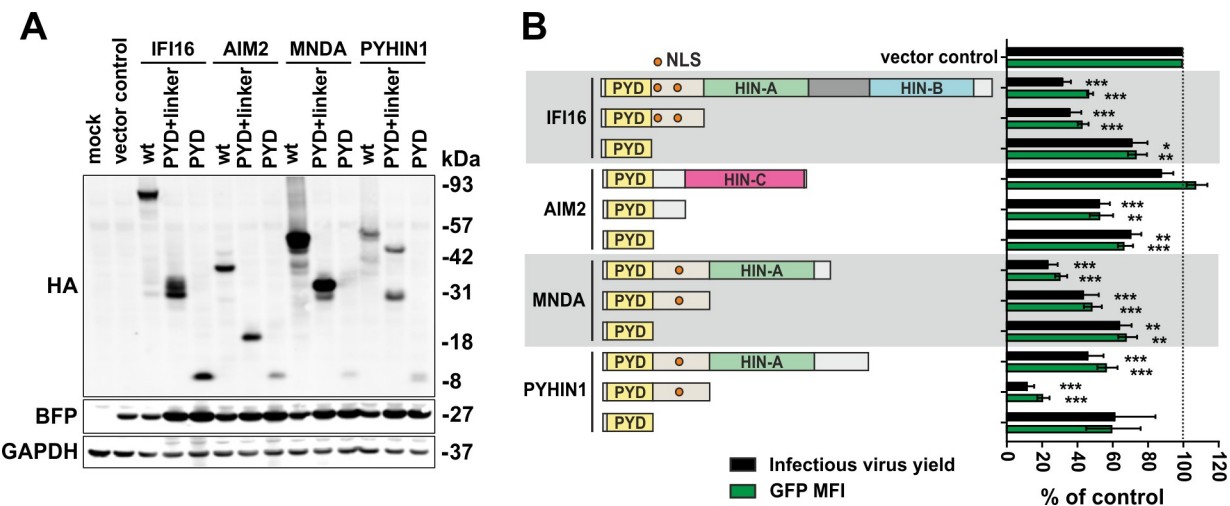

**Fig 5. Determinants of the antiretroviral activity of human PYHIN proteins.** (A) HEK293T cells were transfected with constructs coexpressing the indicated full-length (wt) form of IFI16, PYHIN1, MNDA and AIM2 or just the N-terminal PYD or PYD and linker region and BFP or a vector control and expression was analyzed by Western blot. BFP and GAPDH are used as transfection and loading control, respectively. (B) Effect of mutant PYHIN proteins on infectious virus yield (black) and levels of LTR-dependent eGFP expression (green). HEK293T cells were cotransfected with constructs coexpressing the indicated wt and mutant PYHIN proteins and proviral HIV-1 NL4-3 IRES-eGFP constructs. At 40 hours post-transfection, BFP and eGFP expression was analyzed by flow cytometry. Shown are mean values (±SEM) derived from seven experiments relative to those obtained in the presence of the empty control vector (100%). * p < 0.05, ** p < 0.01, *** p < 0.001.

foci or specks, on which caspase-1 is recruited and autoproteolytically activated [39]. IFI16 has been implicated in the assembly of DNA-sensing inflammasomes in response to nuclear herpesvirus DNA [23,40–42] and HIV DNA [19]. Yet, *bona fide* ASC specks nucleated by IFI16 have not been observed. While some inflammasome sensors bearing a caspase recruitment domain (CARD) can directly recruit and activate pro-caspase-1 in the absence of ASC through CARD-CARD interactions, PYD-containing inflammasomes require the adaptor protein ASC. Efficient maturation of IL-1β has further been shown to require ASC specks [38,43]. To test whether IFI16 can nucleate inflammasome assembly, we reconstituted inflammasomes in a monoclonal HEK293T cell line that was selected to constitutively express ASC-EGFP at low levels with minimal ASC speck background. ASC specks can be detected by the characteristic redistribution of ASC-EGFP fluorescence measured by flow cytometry [44]. While no ASC specks were observed in cells transfected with empty vectors, overexpression of AIM2 (and the simultaneous delivery of expression vectors as potential ligands) led to substantial ASC speck formation (Fig 7A). Overexpression of IFI16 did not induce ASC speck assembly, suggesting that IFI16 is not able to assemble plasmid-DNA induced inflammasomes. To test whether nuclear localization of IFI16 precludes nucleation of ASC specks, we transiently overexpressed IFI16 with the AIM2 linker (cytosolic) as well as AIM2 with the IFI16 linker (nuclear). A clear ASC speck response was initiated by AIM2 with an IFI16 linker, while IFI16 with the AIM2 linker was not able to nucleate inflammasomes (Fig 7A and 7B). Thus, this property is independent of the predominant localization of the PYHIN member. However, AIM2 with the IFI16 linker may nucleate specks while it is transiently located in the cytoplasm. Most importantly, our results indicate that AIM2 is uniquely able to initiate inflammasomes while IFI16 is not.

## Nuclear human PYHIN proteins limit the availability of Sp1

Our finding that the PYD and nuclear localization were generally sufficient for inhibition of HIV-1 suggested that IFI16, PYHIN1 and MNDA act by a common mechanism. To further

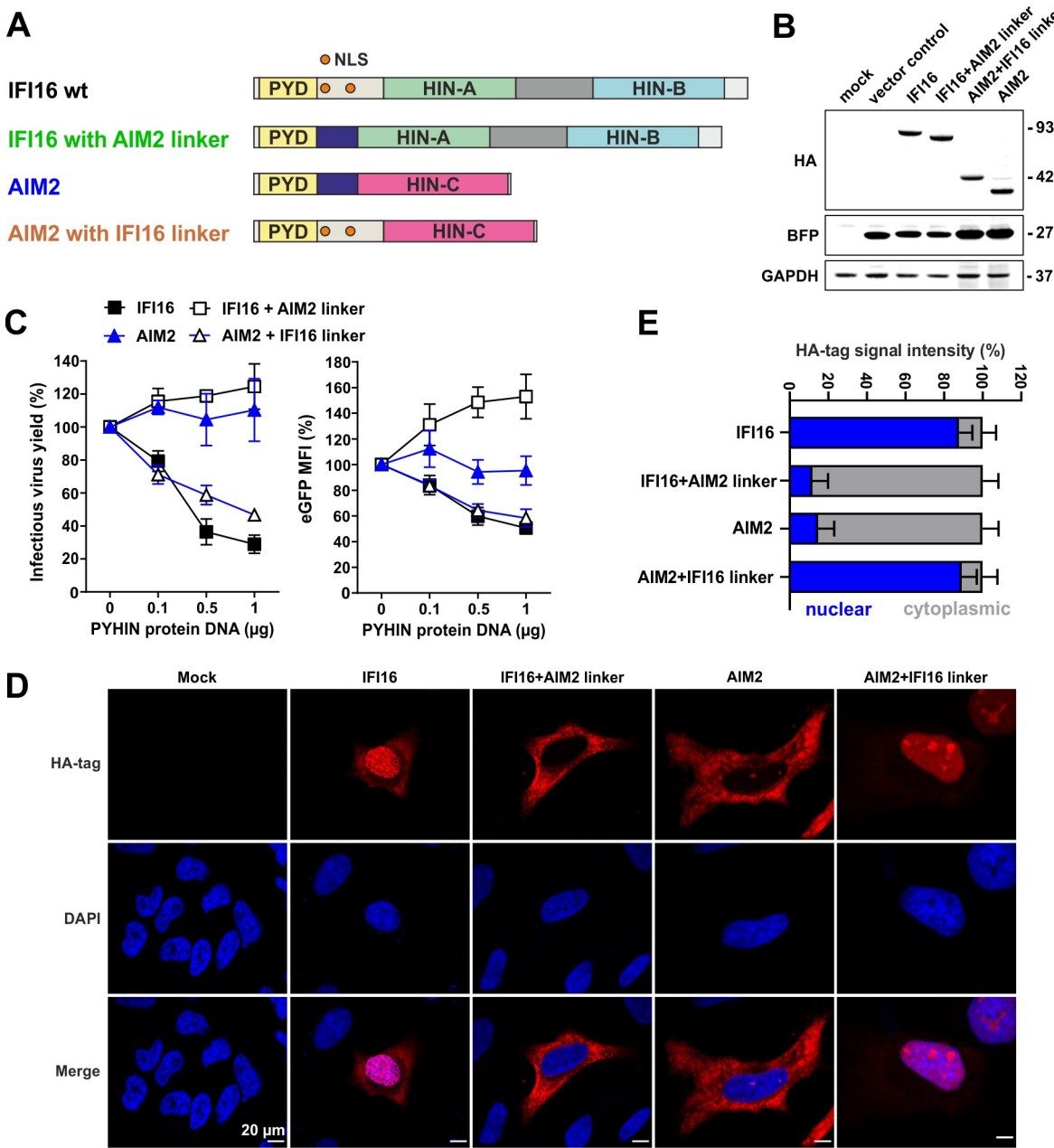

**Fig 6. The NLS of IFI16 confers antiretroviral activity to AIM2.** (A) Schematic presentation of the parental and chimeric IFI16 and AIM2 proteins analyzed. (B) Expression of parental and chimeric IFI16 and AIM2 proteins. HEK293T cells were transfected with IRES-BFP constructs expressing the indicated proteins and examined by Western blot. BFP and GAPDH were used as controls. (C) Antiviral activity of parental and chimeric IFI16 and AIM2 proteins in HEK293T cell. Cells were cotransfected with the indicated PYHIN-IRES-BFP expression vectors and proviral HIV-1 NL4-3 IRES-eGFP constructs. 40 h post transfection, infectious virus yield was determined by infection of TZM-bl cells (left) and LTR-dependent eGFP expression by FACS (right). Mean fluorescence intensities (MFIs) of eGFP expression were measured in the BFP+ population and normalized to the MFI detected in the presence of the empty control vector (100%). Shown are average values (±SEM) from triplicate experiments. (D) HeLa cells were transfected with expression construct for the indicated PYHIN proteins. 48 hours post transfection, cells were fixed and stained for the HA-tag (Cell Signaling). DAPI staining was used to stain DNA. Images were taken using the LSM710 confocal microscope (Carl Zeiss) (63x magnification) and HA signal was quantified with Fiji. n = 3 (30–50 cells/sample ± SD). The scale bar represents 20 μm. (E) Distribution of the indicated proteins in the nucleus and cytoplasm.

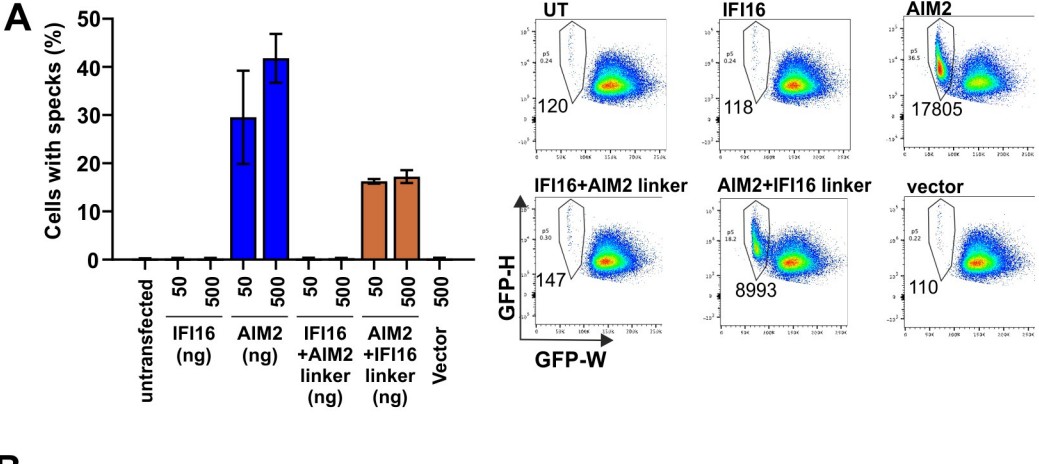

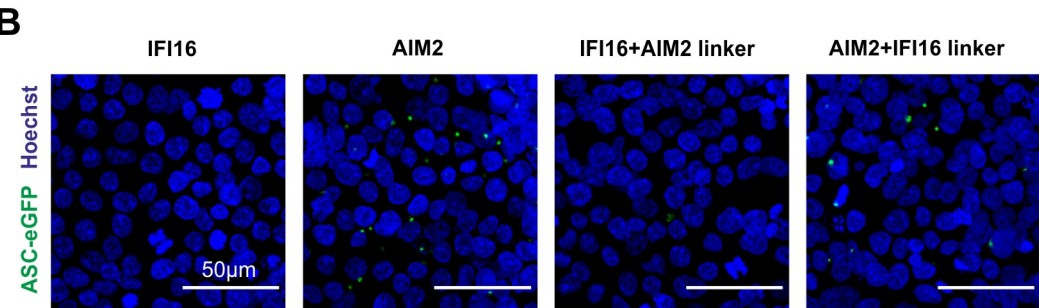

**Fig 7. IFI16 does not nucleate inflammasomes.** (A) HEK293T ASC-EGFP cells were transfected with expression constructs for the indicated PYHIN proteins. 24 hours post transfection, cells were harvested, fixed, and the fraction of cells containing ASC specks quantified by flow cytometry. Average values of cells with ASC specks (±SEM) from three independent experiments (left) and representative FACS plots of HEK293T ASC-EGFP cells transfected with empty vectors, or expressing AIM2 and IFI16 (right). (B) HEK293T ASC-EGFP cells grown on cover slips were treated as described in panel A, fixed, and stained for DNA with Hoechst 33324. Shown are representative images.

examine this, we analyzed seven primary infectious molecular clones (IMCs) of HIV-1 previously shown to differ in their susceptibility to IFI16 [31] and found that they show similar patterns to inhibition by PYHIN1 and MNDA (Fig 8A). The susceptible HIV-1 CH040 and CH058 strains were efficiently inhibited by full-length and PYD plus linker region of IFI16, MNDA and PYHIN1 as well as (less efficiently) by AIM2 (Fig 8B). In contrast, all four human PYHIN proteins displayed little if any inhibitory effect on HIV-1 THRO and CH042. The unexpected exception was the PYD plus linker region truncation mutant of PYHIN1 that was highly active and inhibited even these two HIV-1 IMCs by ≥80% (Fig 8B). Altogether, the results suggest shared determinants of HIV-1 susceptibility to inhibition by different human PYHIN proteins.

Previous data have shown that IFI16 interacts with Sp1 and reduces the availability of this key transcription factor for LTR-driven viral transcription [31]. To determine whether PYHIN1 and MNDA share this mechanism, we performed co-immunoprecipitation experiments in the presence or absence of benzonase that degrades all forms of DNA and RNA (Fig 8C). Using HA-tagged variants of the PYHIN proteins, we found that endogenous Sp1 was efficiently pulled down together with IFI16, MNDA, PYHIN1 and AIM2 (Fig 8D), especially in the absence of competing DNA (Fig 8E). The finding that AIM2 interacts with Sp1 in cellular extracts agrees with our finding that this cytosolic immune sensor acquires more direct antiretroviral activity if its artificially forced into the nucleus (Fig 6A–6D). To further examine the effect of the PYHIN proteins on Sp1 activity, we used a DNA-binding ELISA quantifying

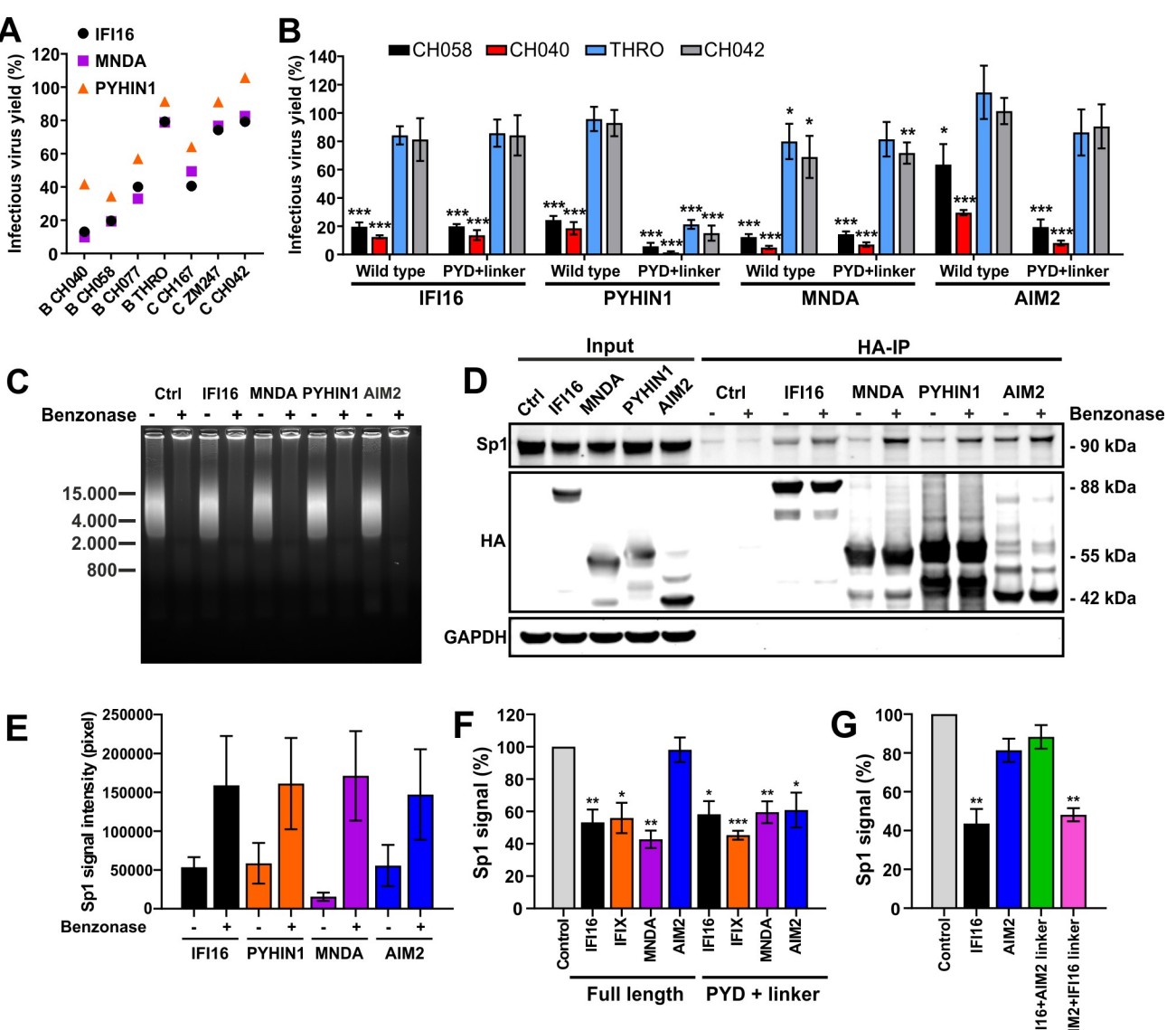

**Fig 8. Human PYHIN proteins interact with Sp1 and reduce its availability.** (A) Inhibition of different primary HIV-1 IMCs by human PYHIN proteins. HEK293T cells were cotransfected with the indicated proviral HIV-1 constructs and a control plasmid or vectors expressing IFI16, PYHIN1 or MNDA. Infectious virus yields in the presence of one PYHIN protein relative to the vector control (100%) was determined by infection of TZM-bl cells. Each symbol represents the average values obtained in three independent experiments. (B) The PYD plus linker region of IFI16, MNDA, PYHIN1 and AIM2 is sufficient for antiretroviral activity. HEK293T cells were transfected and analyzed for infectious virus yield as described in panel A. Shown are average values obtained from four experiments (± SD). * p< 0.05, ** p <0.01, *** p <0.001. (C-E) PYHIN proteins compete with DNA for Sp1 binding. HEK293T cells were transfected with empty vector or expression constructs for the C-terminally HA-tagged PYHIN proteins. 40 hours post transfection, cells were lysed in Co-IP buffer and left untreated or treated with Benzonase before immunoprecipitating the PYHIN proteins using an anti-HA mouse antibody and magnetic beads. (C) Prior to immunoprecipitation, lysates were run on a 1% agarose gel to verify DNA degradation. (D) Proteins were blotted onto PVDF membranes and stained using anti-Sp1 goat, anti-HA rabbit and anti-GAPDH rat antibodies. (E) Mean Sp1 signal intensities (±SD) minus background control measured in three independent experiments. (F, G) HEK293T cells were transfected with expression constructs for the indicated PYHIN protein or their N-terminal PYD and linker regions. 40 hours later, cells were harvested and normalized quantities of nuclear extracts were used for the TransAM Sp1 binding assay (ActiveMotif). Data represent mean values (± SEM) obtained from three experiments.

available Sp1 in nuclear extracts. In agreement with the antiviral activity, IFI16, MNDA and PYHIN1 reduced the levels of Sp1 bound to target DNA sequences while AIM2 had no significant effect (Fig 8F). In all cases, the PYD plus linker region of IFI16 were sufficient to reduce the levels of available Sp1 in nuclear extracts of transfected HEK293T cells. In addition, we

verified that the NLS containing linker region of IFI16 allowed AIM2 to reduce the levels of available Sp1, while the NLS of AIM2 disrupted this effect of IFI16 (Fig 8G).

## Determinants of Sp1 and IFI16 interaction and function

To obtain insights into the determinants of the interaction between Sp1 and the PYDs, we generated a set of seven constructs expressing FLAG-tagged wild type Sp1 or variants containing deletions in its main functional domains (Fig 9A). Mutants were generated based on the UniprotKB database (P08047) and included deletions of the N-terminal region, the transactivation domains, a highly charged region, the Zn-finger domain and the C-terminus. All mutant forms of Sp1 were efficiently expressed (Fig 9B, upper) and co-immunoprecipitated with HA-tagged IFI16

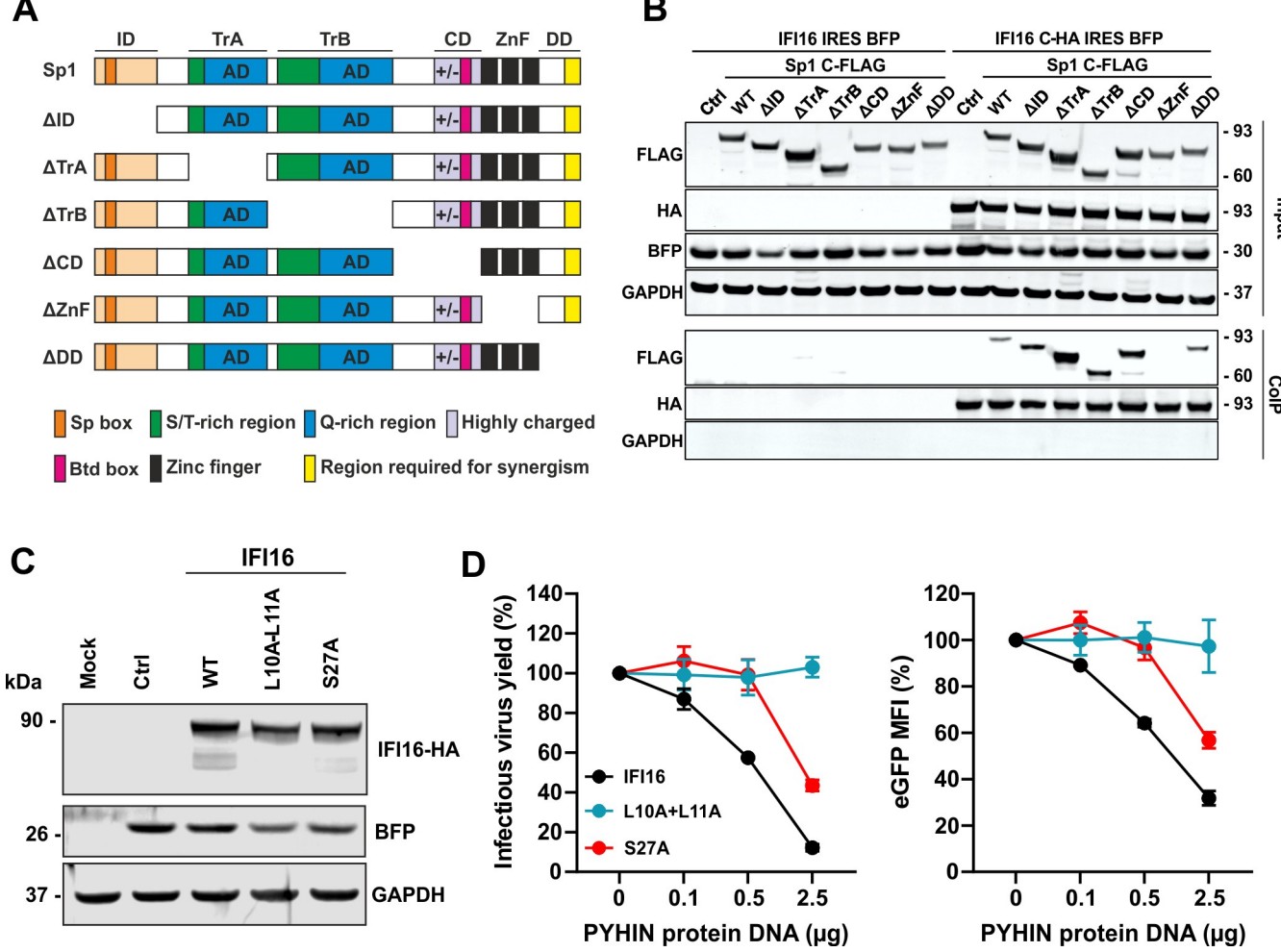

**Fig 9. Determinants of Sp1-IFI16 interaction and antiviral activity.** (A) Schematic overview on mutants Sp1 proteins generated and examined. ID: inhibitory domain; TrA: transactivator domain A; TrB: transactivator domain B; CD: C domain; ZnF: Zinc-fingers; DD:D domain. (B) HEK293T cells were cotransfected with expression constructs for the indicated FLAG-tagged mutant Sp1 proteins and HA-tagged IFI16 or a vector control. At 40 h post-transfection, cells were lysed and IFI16 was immunoprecipitated using anti-HA antibodies and protein A/G magnetic beads. Proteins were blotted and stained using anti-FLAG, anti-HA, anti-BFP and anti-GAPDH antibodies. The upper panels show representative western blots and the lower panels the results of the Co-IP. Results show one representative experiment of three performed. (C) Expression of WT and mutant IFI16 proteins. HEK293T cells were transfected with IRES-BFP constructs expressing the indicated HA-tagged IFI16 proteins. BFP and GAPDH were used as controls. (D) HEK293T cells were cotransfected with a control vector or expression constructs for the indicated IFI16 proteins expressing BFP via an IRES together with a proviral NL4-3-IRES-eGFP construct. Two days later infectious virus yield was quantified by TZM-bl cells infection assay (left) and BFP and eGFP MFI were measured by flow cytometry (right). Mean fluorescence intensity (MFI) of eGFP in the BFP positive population is shown. Shown are mean values (±SD) from four replicates.

(Fig 9B, bottom). The single exception was the Sp1 derivative containing a deletion of the Zn-finger domain that is critical for the ability of this transcription factor to bind DNA (Fig 9B, bottom). To further define the determinants of the antiretroviral activity of IFI16, we introduced mutations of L10A/L11A and S27A into its pyrin domain (S4C Fig). It has previously been reported that these amino acid residues are involved in high affinity protein-protein interactions of the PYD [35]. The mutant IFI16 constructs were both efficiently expressed (Fig 9C). Mutation of the conserved L10/L11 residues at the N-terminus of the PYD fully disrupted the antiviral activity of IFI16, while substitution of S27A resulted in an intermediate phenotype (Fig 9D). Altogether, these results further support that the PYDs of nuclear PYHIN proteins compete with DNA for Sp1 binding and consequently reduce the availability of Sp1 for viral transcription.

## Human PYHIN proteins inhibit HBV, SV40 and LINE-1 elements

Sp1 is critical for efficient activation of many viral promoters and previous studies reported that IFI16 inhibits not only retroviruses [31] but also human cytomegalovirus (HCMV) [26], herpes simplex virus 1 (HSV-1) [28], papillomaviruses [45] and Hepatitis B virus (HBV) [46]. To determine whether AIM2, PYHIN1 and MNDA are also active against diverse viral pathogens, we analyzed their effect on HBV, which causes chronic infections in ~250 million people worldwide [47] and simian virus 40 (SV40), a polyomavirus found in both monkeys and humans. All PYHIN proteins inhibited SV40 infection as seen by a reduced number of cell positive for the Large T antigen, with PYHIN1 showing the most pronounced antiviral effect by reducing the number of infected cells by ~80% (Fig 10A). In contrast, AIM2 and IFI16 were more active against HBV than MNDA and PYHIN1 (Fig 10B). In line with published data on IFI16 [45,46], the differential efficiencies suggested that the antiviral effect of human PYHIN proteins might involve different mechanisms. In agreement with this possibility, the N-terminal PYD plus linker region of IFI16 that is sufficient for antiretroviral activity [31], was inactive against SV40 (Fig 10C). In contrast, mutation of the functional oligomerization residues in the PYD (Oligo) or the NLS (ΔNLS) did not disrupt the activity of IFI16 against SV40 (Fig 10C), although they rendered it inactive against HIV-1 [31]. Unexpectedly, the PYD plus linker mutants of IFI16, PYHIN1, MNDA and AIM2, generally inhibited HBV more efficiently than the corresponding full-length PYHIN proteins, while mutations of IFI16 oligomerization residues or removal of the NLS abrogated its antiviral activity (Fig 10D). Altogether, these preliminary results support that human PYHIN proteins display broad antiviral activity and suggest that virus restriction involves different domains and hence underlying mechanisms.

We have previously shown that IFI16 inhibits transposition and transcriptional activity of LINE-1, the only autonomously active retrotransposon in humans [31]. We found that AIM2, MNDA and PYHIN1 all suppress LINE-1 promoter activity almost as efficiently as IFI16 (Fig 9E) but had little if any inhibitory effect on the CMV promoter (Fig 10F) excluding undesired side-effects on the CMV-driven LINE-1 reporter. Analyses of the efficiency of retrotransposition by a GFP-based reporter assay [48], confirmed that all these human PYHIN proteins suppress retrotransposition about as effectively as IFI16 (Fig 10G). For control, we used active phosphorylation-deficient SAMHD1 (T592A) and an enzymatically inactive mutant (D207N) thereof, lacking anti-LINE-1 activity [48]. The effect of AIM2 came as surprise since our previous analyses showed that the N-terminal PYD plus linker region were required and sufficient for potent inhibition of retrotransposition suggesting that IFI16 restricts HIV-1 and LINE-1 by the same mechanism [31].

## Discussion

In the present study, we show that the ability to interact with the transcription factor Sp1 is a conserved property of the PYDs of human PYHIN proteins. Consequently, IFI16, PYHIN1

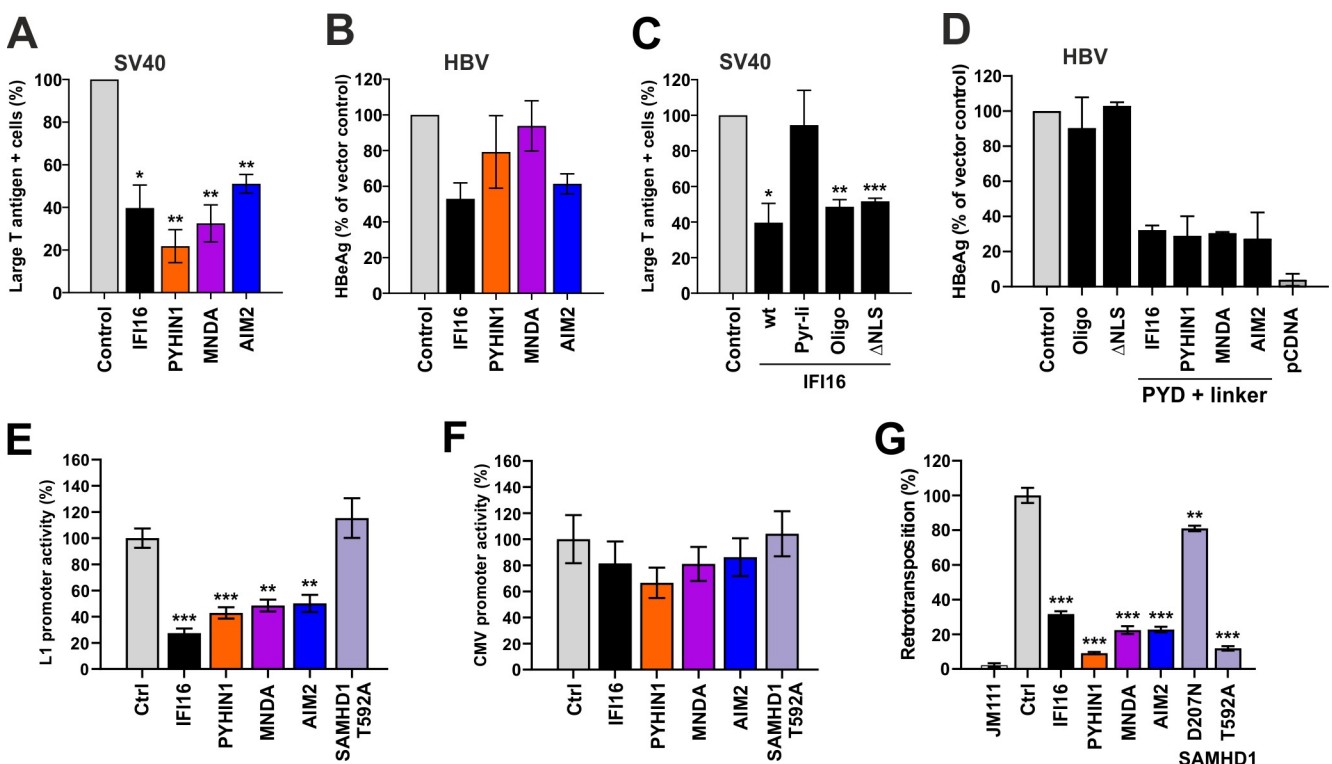

**Fig 10. Effect of human PYHIN proteins on various viruses and LINE-1.** (A, B) Impact of IFI16, PYHIN1, MNDA and AIM2 on (A) SV40 and (B) HBV infection. An empty vector was used as negative control. (C, D) Determinants of the ability of IFI16 or other human PYHIN proteins to inhibit (C) SV40 and (D) HBV. CV-1 and HepG2 cells were transfected with either an empty vector or expression constructs for full length or the indicated mutants of IFI16 and PYHIN proteins. At 24 h post transfection, cells were infected with SV40 or HBV. SV40 infection was evaluated via intracellular staining of the Large T antigen while HBV replication was assessed by quantification of the HBeAg released into the supernatant. Shown are the results from two to three independent experiments ± SD (E, F) HEK293T cells were cotransfected with (C) LINE-1 or (D) CMV promoter constructs driving luciferase gene expression and expression constructs for IFI16 or a control vector. Luciferase activities (relative light units, RLU) in cell lysates were determined two days later (n = 3 ± SEM). (G) HEK293T cells were cotransfected with a retrotransposition-competent (L1) or -defective (L1 neg) LINE 1-GFP reporter plasmid and a vector control or expression constructs for PYHIN proteins or SAMHD1. Five days post-transfection, GFP+ cells were quantified by flow cytometry (n = 3 ± SEM).

and MNDA, which are predominantly localized in the nucleus, share the ability to inhibit HIV-1 by reducing the availability of this transcription factor for proviral transcription. Unexpectedly, the NLS of IFI16 conferred this antiretroviral activity to the well-characterized immune sensor AIM2. This finding supports that the Sp1 interaction represents a fundamental feature of PYDs and agrees with our previous result that antiretroviral activity is conserved between primate IFI16 proteins and their mouse homolog p204 [31]. We have previously shown that IFI16 restricts HIV-1 in macrophages [31]. Here, we demonstrate that this is also the case in CD4+ T cells and further show that PYHIN1 and MNDA cooperate with IFI16 to restrict HIV-1 in macrophages. Notably, Sp1 is not only critical for efficient expression of many viral pathogens but also involved in cell differentiation, growth and apoptosis as well as immune responses and many types of cancers [49,50]. Thus, modulation of the availability of Sp1 for cellular and foreign gene expression by PYHIN proteins might have broad implications that warrant further study.

Our results add to the accumulating evidence that IFI16 restricts various viral pathogens, including herpes-, papilloma-, polyoma- and retroviruses, independently of their effects on the inflammasome and interferon responses [26,27,31,51] and further support that this function is shared by the nuclear human PYHIN proteins PYHIN1 and MNDA. While IFN induction

was not required for HIV-1 restriction, it is noteworthy that IFI16, PYHIN1 and MNDA but not AIM2 significantly boosted type I IFN induction in the presence of STING (Fig 2). It has been reported that IFI16 directly interacts with STING to boost IFN responses in macrophages [13,14,32]. Overexpression of PYHIN1 and MNDA promoted STING dependent IFN induction in transfected HEK293T cells as efficiently as IFI16 (Fig 2) and both factors are efficiently expressed in primary macrophages (Fig 3A). Thus, it is tempting to speculate that, similarly to IFI16, PYHIN1 and MNDA may promote antiviral IFN responses in macrophages. In contrast to IFI16, AIM2 is uniquely able to nucleate ASC specks, even when it is predominantly localized to the nucleus. This suggests that local polymerization of the AIM2 PYD can serve as a template for ASC PYD polymerization [37]. Importantly, overexpression of any confirmed canonical inflammasome sensor is sufficient to nucleate ASC specks by the inherent loss of autoinhibition, ruling out that IFI16 assembles canonical inflammasomes. Nevertheless, this does not rule out indirect roles of IFI16 in promoting the expression of inflammasome components, or as a co-factor for other inflammasome sensors.

Previous studies reported a variety of mechanisms for the antiviral activity of IFI16 against different viral pathogens. For example, studies on herpesviruses proposed that IFI16 inhibits viral promoters through addition of heterochromatin marks or occupation of viral DNAs [24,26]. It has been suggested that IFI16 recognizes foreign DNA in the nucleus because it differs from cellular genomic DNA in its chromatin status [35,52]. Assembly of IFI16 complexes at under-chromatinized foreign DNAs would provide an elegant explanation of how this factor distinguishes friend from foe. Thus, it came as surprise that the HIN domains of IFI16 required for DNA interaction were fully dispensable for its antiretroviral activity [31]. Here, we show that the HIN domain is also dispensable for the antiretroviral activities of MNDA and PYHIN1. We further demonstrate that Sp1 interaction is a conserved property of the PYDs of human PYHIN proteins including the largely cytosolic sensor AIM2. Notably, it has been shown that some amino acid residues in the N-terminal PYD that are critical for high affinity protein-protein interactions are conserved among human PYHIN proteins [35]. These results agree with the ability of human PYHIN proteins to restrict various viral pathogens depending on Sp1 for effective transcription. Other PYD interactions, such as the nucleation of ASC, may require properties specific to distinct PYHIN family members [53].

We have previously shown that subtype C HIV-1 strains are less susceptible to IFI16 inhibition than HIV-1 subtype A, B and D strains in primary macrophages and hardly affected by differences in Sp1 expression levels [31]. In the present study, we demonstrate that IFI16 efficiently restricts the subtype B HIV-1 CH040 and CH058 strains but has no significant effect on the subtype C HIV-1 CH042 strain in primary CD4+ T cells. Altogether, our results show that Sp1 is a limiting factor for efficient transcription of most subtypes of HIV-1 and that the availability of this transcription factor is reduced by nuclear PYHIN proteins. In addition, our data indicate that reduced Sp1 dependency and consequently less susceptibility to inhibition by IFI16 and other nuclear PYHIN proteins are characteristics of primary subtype C HIV-1 strains. Thus, it is tempting to speculate that relative resistance to nuclear PYHIN proteins might contribute to their effective spread in the human population. Notably, the subtype B HIV-1 CH040, CH058 and CH042 IMCs all represent transmitted-founder viruses and were generated directly from patient-derived sequences. It is striking that an ~70% reduction of IFI16 increased their production in primary CD4+ T cells by more than an order of magnitude, clearly suggesting an important role *in vivo*. It remains to be clarified why not all subtypes of HIV-1 evolved reduced Sp1 dependency and resistance against IFI16 like HIV-1 subtype C strains. We have previously shown that the LTR determines Sp1 dependency and IFI16 sensitivity [31]. Currently, we are mapping the specific determinants and investigating whether reduced dependency on Sp1 may come at a cost. IFI16 is expressed at high levels in

resting central memory T cells. Thus, further studies on the roles of IFI16 and Sp1 in the establishment and maintenance of the latent viral reservoirs and potential HIV-1 subtype-specific differences are highly warranted.

The PYHIN protein family shows different degrees of expansion in different mammalian species ranging from one gene in cows, over four in humans to 14 in mice [54]. Notably, genes involved in innate antiviral defense frequently seem to have a history of duplications and functional diversification. For example, duplications in APOBEC3 and TRIM genes that play key roles in antiviral immunity occurred independently in various species [55–57]. In the ever-ongoing virus-host arms race structural diversification and distinct expression patterns, expression of several PYHIN proteins might minimize the risk of effective viral antagonism or evasion and allow functional diversification. Our results suggest that limiting the availability of Sp1 is a fundamental effector mechanism of nuclear PYHIN proteins that might be effective against the large variety of pathogens using this transcription factor for efficient gene expression. However, as mentioned above the sequence diversity of mammalian PYHIN proteins, published data and our own results also suggest they have acquired a variety of effector and sensing mechanisms to restrict invading pathogens. For example, the PYD plus linker regions of IFI16, MNDA, PYHIN1 and AIM2 were all sufficient for restriction of HBV, while deletion of the NLS eliminated the activity of IFI16 against this virus (Fig 10). In contrast, the PYD plus linker region was insufficient and the NLS not required for the inhibitory effect of IFI16 on SV40. In turn, the NLS of IFI16 did not abrogate the function of AIM2 as an inflammasome sensor. These preliminary results suggest that the antiviral function of the four human PYHIN proteins IFI16, AIM2, PYHIN1 and MNDA involves overlapping but distinct mechanisms. The relative importance against various major pathogens remains to be determined and may vary in different cell types or tissues.

One unexpected finding was that the PYD plus linker region fragments were sometimes more effective in restricting HIV-1 (Fig 8B) and HBV (Fig 10D) than the full-length PYHIN proteins suggesting potential inhibitory roles of the HIN domain(s). In agreement with this possibility, it has been proposed that PYHIN proteins may adopt an autoinhibited conformation in which the pyrin domain blocks the DNA-binding surface of the HIN domain [53]. This scenario implies that deletion of the HIN domains might facilitate the interaction of the pyrin domains with cellular factors including Sp1. Interestingly, the PYD plus linker region of PYHIN1 even reduced infectious virus yield of HIV-1 THRO and CH042 by ≥80%, although these two HIV-1 strains were otherwise largely resistant to PYHIN proteins (Fig 8B). Pyrin domains may interact with various cellular factors [58] and it will be interesting to further determine why the PYD of PYHIN1 seems to be particularly effective in restricting these HIV-1 variants. The mechanistic model proposed for PYHIN protein activation mentioned above also involves that upon HIN binding to foreign dsDNA, the PYD is displaced and may interact with its cellular targets [53]. We found, however, that Sp1 interaction is substantially enhanced in the absence of DNA (Fig 8C–8E) suggesting that DNA competes with PYHIN proteins for Sp1 interaction rather than facilitating it.

It has been proposed that defense of the genome against endogenous retroelements might have been an additional evolutionary driver for PYHIN proteins [54]. Indeed, our results show that IFI16, AIM2, PYHIN1 and MNDA all suppress retrotransposition of the long interspersed nuclear element-1 (LINE-1). The latter is of particular interest since this is the only autonomously active retrotransposon in humans and it is estimated that up to 17% of the human genome might be derived from LINE-1 [59]. Control of retrotransposition is relevant in preventing autoimmunity as well as cancer development [60,61]. Recently, it has been shown that increased LINE-1 activity promotes type I IFN expression and age-associated inflammation [62]. Thus, control of LINE-1 activity is critical for genetic stability and prevention of

inflammatory disorders. It has been reported that Sp1 and Sp3 binding motifs are located downstream of the major predicted transcription initiation site of the rat LINE-1 promoter [63]. Although no consensus Sp1 binding sites are found near the transcription initiation site of human LINE-1 elements, several G-rich sequences that might potentially interact with Sp1 are present. Our results clearly suggest that PYHIN proteins may play a role in restricting LINE-1 and potentially other retroelements.

In conclusion, we show that the ability to interact with Sp1 is not only conserved between the pyrin domains of the nuclear PYHIN proteins of IFI16, PYHIN1 and MNDA but even in AIM2, which is evolutionarily and functionally distinct [54]. As previously reported for IFI16 [64], PYHIN1 and MNDA also inhibited SV40 in overexpression assays. In contrast to the antiretroviral activity, however, the PYD plus linker region was insufficient for effective inhibition of SV40. While the relevance of PYHIN1 and MNDA in restricting viral infections *in vivo* remains to be determined, our results clearly demonstrate that they are about as effective as IFI16 in restricting HIV-1 in primary macrophages. Perhaps most importantly, our results clearly demonstrate that IFI16 efficiently restricts primary HIV-1 strains in primary CD4+ T cells. To which extent reduced Sp1 dependency and IFI16 sensitivity alter the replicative and transmission fitness, cell tropism and/or capacity to establish latent infections of subtype C HIV-1 strains remains to be determined. Altogether, our results further support that human PYHIN proteins are broad-based antiviral effectors that inhibit viral pathogens by a variety of non-exclusive effector mechanisms that are independent of inflammasome function and IFN induction. The relative importance of e.g. epigenetic modifications, occupation of viral promoters and direct Sp1 binding needs further study. However, the conservation of the interaction between various PYDs and Sp1 together with the known role of this factor in transcription of various viral pathogens support an important role in innate antiviral immunity.

## Materials and methods

### Ethical statement

Experiments involving human blood, CD4+ T cells or macrophages, were reviewed and approved by the Institutional Review Board (i.e. the Ethics Committee of Ulm University). Individuals and/or their legal guardians provided written informed consent prior to donating blood. All human-derived samples were anonymized before use. The use of established cell lines (HEK293T, TZM-bl, HL116, HepG2 and CV-1 cells) did not require the approval of the Institutional Review Board.

### Cell lines

All cells were cultured at 37˚C in a 5% $CO_2$ atmosphere. HEK293T and TZM-bl cells were maintained in Dulbecco's Modified Eagle Medium (DMEM) supplemented with 10% fetal calf serum (FCS), L-glutamine (2 mM), streptomycin (100 μg/ml) and penicillin (100 U/ml). HEK293T cells [66] were provided and authenticated by the ATCC. TZM-bl cells were provided and authenticated by the NIH AIDS Reagent Program, Division of AIDS, NIAID, NIH from Dr. John C. Kappes, Dr. Xiaoyun Wu and Tranzyme Inc [67]. TZM-bl are derived from HeLa cells, which were isolated from a 30-year-old female. HL116 cells were maintained in DMEM containing 10% FCS, glutamine (2 mM), antibiotics and HAT supplement (sodium hypoxanthine (50 μM), aminopterin (200 nM) and thymidine (8 μM)). HL116 cells stably express an IFN-inducible firefly luciferase reporter gene and are derivatives of the human fibrosarcoma HT1080 cell line established from a 35-year-old male [68]. For the SV40 assays, CV-1 cells were a kind gift from J. Kartenbeck (DKFZ, Heidelberg, Germany). CV-1 cells were

derived from the kidney of an adult male African green monkey (*Chlorocebus aethiops*) [69]. HepG2 cells were cultured in DMEM supplemented with 10% FCS, L-glutamine (2 mM), streptomycin (50 U/ml) and penicillin (50 μg/ml). HepG2 were derived from a liver carcinoma of a 15-years old Caucasian male [70]. HEK293T cells constitutively expressing human ASC fused to eGFP (ASC-eGFP) controlled by the UbC promoter were generated by transduction with lentiviruses produced with vectors pRRL UbCp ASC-eGFP Puro, psPax2 and pMD2.G. Cell clones with single lentivirus insertions were selected and maintained in DMEM with 10% FCS and puromycin (1μg/mL). A single clone was selected for minimal background assembly of ASC specks and was used for all experiments.

## Primary cell cultures

PBMCs from healthy human donors were isolated using lymphocyte separation medium (Biocoll separating solution; Biochrom) or lymphoprep (Stemcell). CD4+ T cells were negatively isolated using the RosetteSep Human CD4+ T Cell Enrichment Cocktail (Stem Cell Technologies) or the EasySep Human Naïve CD4+ T cell Isolation Kit (Stem Cell Technologies) according to the manufacturer's instructions. Primary cells were cultured in RPMI-1640 medium containing 10% FCS, glutamine (2 mM), streptomycin (100 μg/ml), penicillin (100 U/ml) and interleukin 2 (IL-2) (10 ng/ml). Monocyte-derived macrophages were obtained by stimulation of PBMC cultures with 15 ng/ml recombinant human M-CSF (R&D systems) and 10% human serum.

## Expression constructs

Expression vectors for IFI16, AIM2, Sp1, STING and APOL6 were generated as previously described [30,31], those of PYHIN1 and MNDA were a kind gift of Professor M. R. Jakobsen and the p65 expression plasmid was kindly provided by Dr. B. Baumann. PYHIN truncated N-terminal mutants with or without the region containing the NLS and chimeras between AIM2 and IFI16 were generated via overlap extension (OE) PCR. For an easier detection of the expressed proteins a C-terminal HA-tag was introduced via the reverse PCR primer. All constructs were sequenced to validate their correctness. PCR primers are listed in S1 Table.

## Transfections and production of virus stocks

HEK293T cells were transiently transfected using the calcium-phosphate precipitation method. One day before transfection, $5 \times 10^5$ HEK293T cells/well were seeded in 6-well plates in 2 ml medium to obtain a confluence of 50–60% at the time of transfection. For transfection, DNA was mixed with 13 μl 2 M $CaCl_2$ and filled up with water to 100 μl. Afterwards, 100 μl 2 x HBS was added dropwise to this mixture, which was mixed by pipetting ten times and added dropwise to the cells. To generate virus stocks, cells were transfected with proviral constructs (5 μg), in combination with an expression vector for VSV-G when indicated. Prior to spinoculation, virus stocks were concentrated by a 10-fold factor using Amicon Ultra– 15 Centrifugal Filters (Merck Millipore). To test the antiviral effect of different proteins, pCG-based expression constructs were cotransfected with proviral constructs. Whenever different amounts of pCG expression vectors were used within an experiment, empty vector control plasmids were used to keep the total DNA amount for all samples constant. The transfected cells were incubated for 8–16 h before the medium was replaced by fresh supplemented DMEM.

## Viral infectivity

To determine infectious virus yield, 6,000 TZM-bl reporter cells/well were seeded in 96-well plates and infected with cell culture supernatants in triplicates on the following day. Three

days post-infection, cells were lysed and *β-galactosidase* reporter gene expression was determined using the GalScreen Kit (Applied Bioscience) according to the manufacturer's instructions with an Orion microplate luminometer (Berthold).

## Western blot

To determine expression of cellular and viral proteins, cells were washed in PBS and subsequently lysed in Western blot lysis buffer (150 mM NaCl, 50 mM HEPES, 5 mM EDTA, 0.1% NP40, 500 μM $Na_3VO_4$, 500 μM NaF, pH 7.5). Lysates were mixed with 4x Protein Sample Loading Buffer (LI-COR, at a final dilution of 1x) supplemented with 10% β-mercaptoethanol (Sigma Aldrich), heated at 95˚C for 5 min, separated on NuPAGE 4±12% Bis-Tris Gels (Invitrogen) for 90 minutes at 130 V and blotted onto Immobilon-FL PVDF membranes (Merck Millipore or Thermo Fisher). Proteins were stained using primary antibodies directed against IFI16 (Santa Cruz #sc-8023), HA-tag (Abcam #ab18181 or Cell Signaling #3724), PYHIN1 (ProSci #61–908), MNDA (Santa Cruz #sc-390739), AIM2 (Santa Cruz #sc-515514), GFP/BFP (Abcam #ab290) GAPDH (BioLegend #607902 or Santa Cruz #sc-365062), Sp1 (Abcam #157123 or Abcam #227383) and Infrared Dye labeled secondary antibodies (LI-COR IRDye). For the detection of endogenous PYHIN1 in monocytes-derived-macrophages (MDMs) transfected with PYHIN1-specific siRNA (see below) a variant of the Signal Enhancer HIKARI for Western Blotting and ELISA protocol kit was used. Briefly, the PYHIN1 primary antibody was diluted 1:1000 in 6 ml of Solution A and incubated with the membrane overnight on a roller mixer (Phoenix Instrument, RS-TR05) at 4˚C. The respective secondary antibody was diluted 1:20.000 in 3 ml of Solution B and incubated for 1 h at room temperature. Proteins were detected using a LI-COR Odyssey scanner and band intensities were quantified using LI-COR Image Studio Lite Version 3.1. Values were normalized to the corresponding GAPDH levels.

## IFN release assay

HL116 cells were used to determine the release of bioactive interferon (IFN) from HEK293T and THP-1 cells. 20,000 HL116 cells/well were seeded in a 96-well plate and stimulated in triplicates with 100 μl cell culture supernatants on the following day. As positive control, supernatants from HEK293T cells infected with Sendai virus. 8 h after stimulation, cells were lysed and firefly luciferase activity was determined using the Luciferase Assay Kit (Promega) according to the manufacturer's instructions with an Orion microplate luminometer (Berthold).

## qRT-PCR

Viral transcript levels were determined in HEK293T cells cotransfected with proviral constructs of NL4-3 (2.5 μg) and increasing doses of expression constructs for IFI16, PYHIN1, MNDA and AIM2 or in $CD4^+$ T cells transduced with VSV-G pseudotyped HIV-1 primary strains. 40 h post-transfection cells were washed with PBS and lysed in RLT Plus buffer containing 1% β-mercaptoethanol. Total RNA was isolated using the RNeasy Plus Mini Kit (QIAGEN) according to the manufacturer's instructions. Residual genomic DNA was removed from the RNA preparations using the DNA-free DNA Removal Kit (ThermoFisher). RNA concentrations were determined using the NanoDrop 2000 Spectrophotometer and for each sample equal RNA amounts were subjected to cDNA synthesis using the PrimeScript RT Reagent Kit (TAKARA) with random 6-mers and oligo dT primers. Reactions without reverse transcriptase were included as controls to exclude contaminations with genomic DNA. cDNA was used for qRT-PCR using TaqMan Fast Universal PCR Master Mix (ThermoFisher) and viral primer/probe sets (Biomers/TIB Molbiol) in multiplex reactions with GAPDH (ThermoFisher) as control. Viral primers and probes were designed as previously described [65] to

measure R-U5/Gag mRNA as indicator of proximal elongation, *nef* mRNA as indicator of transcription that has proceeded almost to the 3'LTR and U3-polyA mRNA as indicator for completed transcription. Primers and fluorescent probes are listed in S2 Table.

## Differentiation, siRNA transfection and infection of macrophages

MDMs were obtained by stimulation of PBMC cultures with 15 ng/ml recombinant human M-CSF (R&D systems) and 10% human AB serum (Sigma Aldrich) in DMEM supplemented with glutamine (2 mM), streptomycin (100 μg/ml) and penicillin (100 U/ml) for 6 days. On days 7 and 9 of differentiation, MDMs were transfected with either a non-targeting control (Eurofins (UUCUCCGAACGUGUCACGUdTdT)) siRNA or IFI16-sepcific (Dharmacon, ON-TARGETplus IFI16 siRNA, #LQ-020004-00-0005), MNDA-specific (Dharmacon, ON-TARGETplus MNDA siRNA, #LQ-010525-00-0005), PYHIN1-specific (Eurofins (GCAACC-GUCUCACAGCUAAdTdT)) and AIM2-specific (Eurofins (UAUGGUGCUAUGAACUC CA-GAUGUCdtdt and UUUCAGCUUGACUUAGUGGCUUUGGdtdt), as published by [71]). siRNA transfections were performed in 12-well plates with two technical replicates for each sample using the Lipofectamine RNAiMAX transfection reagent (Thermo Fisher). Prior to transfection, the medium was replaced by 500 μl fresh supplemented DMEM. For one well, 1.46 μl siRNA (20 μM) and 3 μl Lipofectamine RNAiMAX were mixed with 75 μl Opti-MEM, each. These two solutions were then mixed, incubated at room temperature for 15 min and added dropwise to the cells. 14 h after transfection on day 7 of differentiation, the medium containing the transfection mix was replaced by fresh supplemented DMEM. 14 h after transfection on day 9 of differentiation, cells were infected with 300 μl of HIV-1 AD8 diluted in 500 μl of DMEM containing 10% FCS, glutamine (2 mM), streptomycin (100 μg/ml), penicillin (100 U/ml). 12 h after infection, the input virus was removed by washing once with 1 ml PBS and 1 ml fresh DMEM containing 10% FCS, glutamine (2 mM), streptomycin (100 μg/ml), penicillin (100 U/ml) was added to the cells. Infectious virus yield was determined in cell culture supernatants harvested 3 and 6 days after infection. To monitor knock-down efficiencies by Western blot, cells were washed with 1 ml PBS and directly lysed in the plate 6 days after infection.

## Transfection and transduction of CD4+ T lymphocytes

CD4+ T lymphocytes were isolated from healthy donors as described above. Cells were stimulated with IL-2 (10 ng/ml) (MACS Miltenyi Biotec) and with anti-CD3/CD28 beads (Life Technologies). Cells were cultured in RPMI-1640 medium containing 20% FCS. Three days later, $6*10^6$ cells were transfected with 1.5 μg of shRNA mix against either a control sequence or IFI16 (Origene) using the Amaxa Human T Cell Kit (Lonza), U15 program. Alternatively, $1*10^6$ cells were transfected with the HiFi Cas9 Nuclease V3 (IDT)/gRNA complex (80 pmol/300 pml) (Lonza) using a non-targeting (IDT (ACGGAGGCTAAGCGTCGCAA)) or an IFI16-specific (IDT (GACCAGCCCTATCAAGAAAG)) sgRNAs, using the Amaxa 4D-Nucleofector Human Activated T Cell P3 Lonza Kit (Lonza), pulse code EO115. At three- and four-days post shRNA and Cas9/sgRNA-transfection respectively, $3*10^5$ cells/sample were transduced with the indicated VSV-G pseudotyped HIV-1 strains by spinoculation. After three days, supernatants were harvested, diluted 1:10 in DMEM containing 10% FCS, L-glutamine (2 mM), streptomycin (100 μg/ml) and penicillin (100 U/ml), and used to quantify viral capsid protein p24 via ELISA, as previously described [72], and infectious virus yield via the TZM-bl reporter cells assay.

## Proximity ligation assay

The Proximity Ligation assay was performed as previously described [73]. Three days post-isolation, CD4+ T lymphocytes were fixed and stained over-night using a rabbit anti-Sp1 (Abcam

#227383), a mouse anti-IFI16 (Santa Cruz, sc-20800) and a mouse anti-HA tag (Cell Signaling, #2999) antibodies. Given the high cross-reactivity of the PYHIN1 antibody with IFI16, CD4[+] T lymphocytes were transfected with an HA-tagged variant of PYHIN1 prior to fixation and staining.

### Flow cytometry

Flow cytometry was used to determine the effect of PYHIN proteins on HIV-1 LTR-driven eGFP expression and cell viability. HEK293T cells were cotransfected with expression constructs for PYHIN proteins co-expressing BFP via an IRES and HIV-1 NL4-3 proviral constructs co-expressing eGFP via an IRES. 48 h after transfection cells were harvested, washed in PBS with 1% FCS and fixed in 2% PFA for 30 min at 4°C. Mean fluorescence intensities (MFI) of eGFP in the BFP+/eGFP+ population and percentage of eGFP+ cells in the BFP+ population (P2/(P2+P1)) was determined. To assess any PYHIN-mediated cytotoxic effect, 48 hours after transfection cells were gently washed in PBS, trypsinized (Pan Biotech, Trypsin/EDTA 0.05%/ 0.02% in PBS w/o Ca, Mg) and stained with the eBioscience Fixable Viability Dye eFluor 780 (Thermo Fisher, #65-0865-14) for 15 minutes at room temperature at dark. Cells were then washed in PBS with 1% FCS and fixed in 2% PFA as previously described and the percentage of dead and living cells was determined on the whole population. Flow cytometric measurements were performed using a BD FACS Canto II flow cytometer.

### Inflammasome assay

To quantify assembly of PYHIN protein-nucleated ASC specks, HEK293T ASC-eGFP cells were seeded into 24-well plates and transfected with 50 or 500 ng of empty vector or expression constructs for the indicated PYHIN proteins (wild type or chimeras) using Lipofectamine 2000. 24 h post transfections, cells were washed with PBS, trypsinized, fixed in 4% formaldehyde, and resuspended in FACS buffer (2% FCS, 5 mM EDTA, 0.02% NaN$_3$ in PBS). Flow cytometric measurements were performed using a BD FACS Canto II flow cytometer. Height and width of the eGFP signal was measured to identify and quantify cells with ASC specks (low width, and high intensity of eGFP signal).

### SV40 production and infection assays

SV40 propagation was carried out as described previously [74]. To investigate the potential antiviral effect of PYHIN proteins against SV40, 1x10[5] CV-1 cells were seeded in 12-well plates in DMEM with 10% fetal calf serum (FCS) about 16 h prior to experimentation. CV-1 cells were transiently transfected with either a control vector or expression constructs for wild type and mutant PYHIN proteins co-expressing BFP via an IRES using Lipofectamine 2000 according to the manufacturer's protocol and subsequently infected with SV40 at a MOI 5 (PFU/cell) at 24 h post transfection. At 24 h p.i., cells were immunostained with an SV40 T-Ag antibody (Santa Cruz, sc-20800) and a goat anti-rabbit Alexa Fluor 488 coupled antibody (Molecular Probes, A11034), and infectivity was scored by flow cytometry (Guava easyCyte, Millipore).

### HBV-infection assays

HepG2 cells were cultured in DMEM medium supplemented with 10% fetal calf serum, 2 mM L-glutamine, 50 U/mL penicillin and 50 μg/mL streptomycin. For HBV infection, 3x10[5] HepG2 cells were seeded in a 24-well-plate (0.5 ml per well). The following day, cells were transfected with 0.25 μg of pXL4-hNTCP plasmid encoding human NTCP and 0.25 μg of PYHIN expression vectors using Mirus TransIT LT1 transfection reagent (Mirus, Germany).

24 h after transfection, cells were washed twice with PBS and then inoculated with HBV in a medium containing 2.5% dimethyl sulfoxide (DMSO) and 4% polyethylene glycol (PEG) 8000 (Sigma). One day after infection, cells were washed with PBS twice and further cultured in medium containing 2.5% DMSO and refreshed every second day. The culture medium between day 4 to 6 post infection were measured for HBeAg with Architect assay (Abbott, Germany).

## Microscopy

Confocal immunofluorescence microscopy was used to determine the subcellular localization of the AIM2, IFI16 and the respective NLS chimeras. 40,000 HeLa cells/well were seeded on 13 mm diameter glass cover slips previously UV-sterilized 24-well plates. On the following day, cells were transfected with expression constructs for the proteins of interest (1 μg) using the polyethylenimine (PEI, in house made solution). Briefly, PEI (10 μl) and DNA were added into two separate OptiMEM solutions which were then mixed and incubated at room temperature for 20 minutes. The resulting master mixes were then added dropwise to the cells and after 6 hours the medium was replaced with fresh DMEM. 48 hours after transfection, cells were washed with ice-cold PBS, fixed in 4% PFA for 20 min at room temperature, permeabilized and blocked in PBS 0.5% Triton X-100 5% FCS for 30 min at room temperature. Overexpressed proteins were stained using an anti-HA (Cell Signaling, #3724) and AF-488 (Invitrogen, #A-11008), while the nuclear DNA was stained using Hoechst *33342 (Invitrogen, #62249)*. Cover slips were mounted on glass slides using Mowiol mounting medium and dried overnight at 4˚C. Confocal microscopy was performed using an LSM710 confocal microscope (Carl Zeiss) and HA-tag signal intensities in betwen 30 and 50 cells per sample were quantified using Fiji image processing software (ImageJ).

## Viral promoter activity

To determine the effect of PYHIN proteins, Sp1 and p65 on the activity of different viral promoters, 22,000 HEK293T cells/well were seeded in poly-L-lysine-coated 96-well plates and transfected in triplicates using the calcium-phosphate transfection method. Cells were cotransfected with firefly luciferase reporter constructs under the control of the HIV-1 LTR (0.3 μg) or the CMV IE promoter (2 μg) and either expression constructs for PYHIN proteins (50 ng), increasing doses of an expression construct for Sp1 or p65, or a vector control. When indicated, an expression construct for HIV-1 NL4-3 Tat (500 pg) was cotransfected to activate the LTR promoter. 40 h post-transfection, cells were lysed and firefly luciferase activity was determined using the Luciferase Assay Kit (Promega) according to the manufacturer's instructions with an Orion microplate luminometer (Berthold).

## Coimmunoprecipitation assay

To investigate the interaction between Sp1 and PYHIN proteins, HEK293T cells were transfected with expression constructs for HA-tagged PYHIN proteins or a vector control (1 μg). 48 h after transfection, cells were lysed in Western blot lysis buffer and PYHIN proteins were immunoprecipitated using anti-HA antibodies (Abcam, #ab18181) and Pierce Protein A/G Magnetic Beads (ThermoFisher). To examine the effect of nucleic acids on the interaction between PYHIN proteins and Sp1, cell lysates were treated with benzonase (750 U/ml) in the presence of $MgCl_2$ (1 mM) for 2 hours at room temperature before adding anti-HA antibodies. Beads were incubated for 1 h at room temperature and washed three times with 1 ml NP40 wash buffer (50 mM HEPES, 300 mM NaCl, 0.5% NP40, pH 7.4) before incubation with 60 μl 1 x Protein Sample Loading Buffer (LI-COR) at 95˚C for 10 minutes to remove bound proteins. After addition of 1.75 μl β-mercaptoethanol the eluate was used for Western blotting. Proteins were stained using goat anti-Sp1 (Abcam, #ab157123), rabbit anti-HA (Cell Signaling,

#3724) and mouse anti-GAPDH (Santa Cruz, #sc-365062) antibodies. To assess the interaction between IFI16 and the C-FLAG tagged Sp1 mutants, samples were initially pre-cleaned with beads for 1 h at 4°C before immunoprecipitating IFI16 using an anti-HA tag mouse antibody (Abcam #18181) overnight at 4°C and. Beads were subsequently incubated for 4 h at 4°C prior to the three washing steps. Proteins were stained using a mouse anti-FLAG (Sigma Aldrich, #F1804), rabbit anti-HA (Cell Signaling, #3724), rabbit anti-GFP/BFP (Abcam, #ab290) and rat anti-GAPDH (Biolegend, #607902) antibodies. To confirm removal of nucleic acids, 20 µl of lysates were diluted 1:5 with Roti-Load DNA (with glycerol) (Roth, #X904.1) and run on a 1% agarose gel (VWR-Peqlab, #9012-36-6) supplemented with 1 drop of ethidium bromide (ITW Reagents, #1239-45-8) for 25 minutes. Gel pictures were taken using a Universal Hood II Gel Doc System (Biorad, #721BR01318) and processed with ImageJ. To validate the interaction between Sp1 and PYHIN proteins in CD4+ T lymphocytes, cells were isolated and activated as previously described. At three days post-isolation, cells were lysed in Co-IP lysis buffer (variant of the Western blot lysis buffer containing 0.2% Triton-X and 0.3% NP40), protein concentration was assessed using the NanoDrop 2000 and a total of 3.000 µg of protein per samples were used. Endogenous Sp1 was immunoprecipitated using a rabbit anti-Sp1 (Abcam, #227383) pre-incubated with beads over-night at 4°C for 4 h at 4°C.

## Sp1-DNA binding assay

HEK293T cells were transfected with expression constructs for wild type or N-terminally truncated PYHIN proteins (1 µg). 48 h post transfection, cells were harvested and nuclear proteins were isolated. Cells were washed with PBS and collected in 1 ml ice-cold Phosphatase Inhibitor Buffer (6.25 µM NaF, 12.5 mM β-glycerophosphate, 1.25 mM NaVO₃, in PBS). Cells were pelleted by centrifugation at 300 x g for 5 minutes at 4°C, resuspended in 1 ml Hypotonic Buffer (20 mM HEPES, 5 mM NaF, 0.1 mM EDTA, pH 7.5) and incubated on ice for 15 min to allow swelling of the cells. After addition of 50 µl 10% NP40, the homogenate was centrifuged at 20,800 x g for 1 min at 4°C. The supernatant was removed and the pellet was resuspended in 50 µl Complete Lysis Buffer AM1 (ActiveMotif). Tubes were incubated for 30 min at 4°C with end-over-end mixing, centrifuged at 20,800 x g for 10 min at 4°C and the supernatants (nuclear cell extract) was transferred to fresh tubes. Protein concentrations were determined using the Pierce Gold BCA Protein Assay Kit (ThermoFisher) according to the manufacturer's instructions. Volumes corresponding to 40 µg of nuclear extracts from samples with endogenous Sp1 levels were used to determine Sp1 binding to consensus target DNA sequences using the TransAM Sp1 Transcription Factor Assay Kit (ActiveMotif) according to the manufacturer's instructions.

## Statistical analyses

Statistical analyses were performed using GraphPad PRISM 8 (GraphPad Software). P-values were determined using a two-tailed Student's t test. Quantification of the p24 concentration in the supernatants via ELISA was achieved with GraphPad PRISM 8, using the "Sigmoidal, 4PL, X is log(concentration)" analysis option. Unless otherwise stated, data are shown as the mean of at least three independent experiments ± SD. Significant differences are indicated as: *, $p < 0.05$; **, $p < 0.01$; ***, $p < 0.001$. Statistical parameters are specified in the figure legends.

## Supporting information

**S1 Fig. Inducibility of the CMV-IE and HIV-1 LTR promoters by Sp1 and p65 NF-κB.** (A, B) HEK293T cells were cotransfected with luciferase reporter constructs under the control of the HIV-1 LTR (0.3 µg) or the CMV IE promoter (2 µg) and either an expression vector for (A) Sp1 or (B) p65, or a vector control. 48 hours post-transfection, luciferase activities were

determined (n = 4 ±SD). $^*$ p $<$ 0.05, $^{**}$ p $<$ 0.01, $^{***}$ p $<$ 0.001.
(TIF)

**S2 Fig. Inducibility of human PYHIN proteins and knock-down in macrophages.** (A) CD4+ T lymphocytes or macrophages treated and analyzed by Western blot as described in the legend to Fig 3A. The bars show mean IFI16, PYHIN1, MNDA and AIM2 levels of three to five donors ±SEM. Calculations of the levels of PYHIN protein expression were always normalized to the GAPDH control. $^*$ p$<$0.05, $^{**}$ p$<$0.01. (B) Western blot analysis of human MDM treated with PYHIN-specific or control siRNA. Shown is a representative example of three donors. Quantitative analyses are provided in Fig 3B.
(TIF)

**S3 Fig. Effect of IFI16 shRNA knock-down on HIV-1 production and transcription.** (A-D) CD4+ T cells were isolated, activated with IL-2 and anti-CD3/CD28 beads, treated with a mix of a control or an IFI16-targeting shRNA and transduced with the VSV-G pseudo-typed HIV-1 strains and infectious virus yield was assessed 72 hours later. Infectious virus yields (A), p24 antigen production (B), the levels of viral RNA transcripts (C) and IFI16 expression levels (D) were determined three days post-transduction. Numbers above bars indicate n-fold change between cells treated with control or IFI16 specific gRNA.
(TIF)

**S4 Fig. Features of the PYD sequences of human PYHIN proteins.** (A) HEK293T cells were cotransfected with HIV-1 NL4-3-IRES-eGFP and expression constructs for full length or mutants forms of PYHIN proteins. At 48 hours post transfection, cells were processed for FACS analysis and analyzed for eGFP and BFP expression. Numbers indicate eGFP MFI in the BFP+eGFP+ population. (B) Expression of PYHIN proteins does not cause cytotoxic effects. HEK293T cells were transfected with an empty vector or expression constructs for the indicated factors, harvested 48 hours later and stained with the Fixable Viability Dye eFluor 450 for flow cytometry. The living/dead population was assessed via FACS (n = 2–3 ± SD). A construct expressing APOL6 was used as a positive control. (C) Amino acid alignment of the N-terminal region of IFI16, PYHIN1, MNDA and AIM2. The shaded area highlights the PYDs, dots indicate amino acid identity and dashes gaps.
(TIF)

**S1 Table. Primers used to generate pCG_IRES_BFP expression constructs.**
(DOCX)

**S2 Table. Primers and probes used for qRT-PCR.**
(DOCX)

## Acknowledgments

We thank Martha Mayer and Daniela Krnavek for technical assistance. TZM-bl cells were obtained through the NIH AIDS Reagent Program, Division of AIDS, NIAID, NIH: TZM-bl from Dr. John C. Kappes, Dr. Xiaoyun Wu and Tranzyme Inc. psPAX2 and pMD2.G were gifts from Didier Trono (Addgene plasmids #12260 and # 12259). We also like to acknowledge the assistance of the Flow Cytometry and the Microscopy Core Facility at the Medical Faculty, University of Bonn.

## Author Contributions

**Conceptualization:** Matteo Bosso, Yi Ni, Stephan Urban, Mario Schelhaas, Florian I. Schmidt, Thomas Gramberg, Daniel Sauter, Frank Kirchhoff.

**Funding acquisition:** Mario Schelhaas, Florian I. Schmidt, Thomas Gramberg, Konstantin M. J. Sparrer, Daniel Sauter, Frank Kirchhoff.

**Investigation:** Matteo Bosso, Caterina Prelli Bozzo, Dominik Hotter, Meta Volcic, Christina M. Stürzel, Annika Rammelt, Yi Ni, Miriam Becker, Sabine Wittmann, Maria H. Christensen.

**Methodology:** Matteo Bosso, Caterina Prelli Bozzo, Dominik Hotter, Meta Volcic, Konstantin M. J. Sparrer.

**Resources:** Stephan Urban, Konstantin M. J. Sparrer, Frank Kirchhoff.

**Supervision:** Dominik Hotter, Daniel Sauter, Frank Kirchhoff.

**Writing – original draft:** Frank Kirchhoff.

**Writing – review & editing:** Matteo Bosso, Dominik Hotter, Mario Schelhaas, Florian I. Schmidt, Thomas Gramberg, Konstantin M. J. Sparrer, Daniel Sauter, Frank Kirchhoff.

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
