## [Decision Letter · Decision Letter 0]

6 Apr 2020

Dear Dr. Kirchhoff,

Thank you very much for submitting your manuscript "Nuclear human PYHIN proteins exert broad antiviral activity by inhibiting the host transcription factor Sp1" for consideration at PLOS Pathogens. As with all papers reviewed by the journal, your manuscript was reviewed by members of the editorial board and by several independent reviewers. Two of the reviewers felt that this manuscript was not a sufficient advance on previous, related work from your laboratory (I am unclear on whether reviewer 1 was aware of this work) and all three felt that there were significant technical problems with the data, in particular due to the extensive use of overexpression experiments in non-physiological 293T cells. In light of these reviews (below this email), we would like to invite the resubmission of a significantly-revised version that takes into account the reviewers' comments. This will be sent back to the same reviewers so they can decide whether these concerns have been adequately addressed. Therefore, I cannot guarantee that this manuscript will be found to be acceptable for publication even after revision and you may wish to consider submission elsewhere.

Sincerely,

Bryan R. Cullen

Associate Editor

PLOS Pathogens

Thomas Hope

Section Editor

PLOS Pathogens

Kasturi Haldar

Editor-in-Chief

PLOS Pathogens

orcid.org/0000-0001-5065-158X

Michael Malim

Editor-in-Chief

PLOS Pathogens

orcid.org/0000-0002-7699-2064

Reviewer's Responses to Questions

**Part I - Summary**

Reviewer #1: In this study Bosso and colleagues have investigated the antiviral activities of PYHIN proteins with a focus on the less well-characterized proteins MNDA and PYHIN1. The authors provide evidence for a model in which these proteins have broad antiviral activity through interaction with, and perhaps sequestration of, the transcription factor Sp1. The experiments are carefully thought out and performed and the data are in the most part compelling. If the Sp1 interaction business can be sorted out satisfactorily this will be a comprehensive study with good impact.

Reviewer #2: The overall quality of this study is poor, relying largely on inhibition of HIV-1 production by overexpressed Pyhin proteins. Several experiments are unconvincing (see below). The study is also not particularly novel, as the same group recently published (in a significantly stronger paper) that one of this group of pyhin proteins inhibits HIV-1, via the same mechanism (Sp1 sequestration) as reported here

Reviewer #3: The authors reported in a recent paper that IFI16, an interferon-inducible member of the family of pyrin and HIN domain containing (PYHIN) proteins, “restricts HIV-1 independently of immune sensing by binding and inhibiting the host transcription factor SP1”. They also reported that SP1 binding and anti-HIV-1 activity depend on the IFI16 pyrin domain (PYD) and its nuclear localization, but not on the DNA-binding HIN domain.

In the present study, the authors extend these findings by showing that the PYDs of other human PYHIN proteins also bind SP1 and inhibit HIV-1, as long as they are localized to the nucleus. Additionally, native human PYHIN proteins MDNA and PYHIN1 but not AIM2 are shown to be nuclear and to inhibit HIV transcription when overexpressed in 293T cells. Furthermore, knockdown experiments indicate that PYHIN1 and MNDA restrict HIV-1 in primary human macrophages. Altogether, the evidence presented suggests that nuclear PYDs, and in some cases the native proteins, inhibit HIV and some other viruses, as well as retrotransposons, by reducing the availability of the SP1 transcription factor. While this is an interesting model, most of the data on HIV were obtained with a rather artificial experiment setup, and the manuscript would benefit from additional evidence establishing relevance of the observations in more physiologically relevant HIV target cells.

**Part II – Major Issues: Key Experiments Required for Acceptance**

Reviewer #1: 1. A key experiment presented in Fig7 is the interaction between IFI16 and Sp1 full length and deletions by co-immunoprecipitation. Unfortunately the Sp1 protein and most of its fragments clearly bind the beads in the absence of HA-Tagged IFI16. This is problematic because the pattern of binding with and without the immunoprecipitated IFI16 is almost identical. Note that the deltaZNF Sp1 fragment doesn’t bind the beads and doesn’t co-precipitate with IFI16. The only specific interaction appears to be the deltaCD Sp1 fragment, which is only present in the co-precipitate, and not in the absence of tagged IFI16 but the authors don’t specifically discuss this. However, given this band is the weakest co-precipitated protein, this may simply be below the detection limit in the absence of IFI16. Basically, this experiment is not compelling enough to hang the whole story on including the title of the paper.

It is true that the precipitated bands are denser but the BFP bands are slightly denser on the right. Unfortunately the specificity is pattern pretty much identical with and without the HA-tagged IFI16.

I’m not sure how to resolve this. The authors might take this out and revise their conclusions. They might sort the conditions out so the co-immunoprecipitation is specific. Unfortunately, if the protein you’re working on binds the beads, then one can’t really use this technique and conditions to measure protein-protein interactions.

2. bottom page 10. Do I understand correctly that the authors think that AIM2 can activate ASC specks in the nucleus when fused with the IFI16 NLS. I may have misunderstood this but they say that the AIM2-IFI16 chimera goes into the nucleus and still activates ASC speck formation. I would interpret this as AIM2 activating ASC specks in the cytoplasm as usual due to remaining cytoplasmic AIM2-IFI16 chimera or newly synthesized AIM2-IFI16 before it goes into the nucleus. The discussion on line 317 seems to imply that in this case the ASC specks might be in the nucleus, which I doubt. If they think the specks might be in the nucleus in this case, they could easily test that by IF, which they should do if that’s what they think. Please clarify the interpretation of this chimera result. No more experiments are necessarily required for this point.

Reviewer #2: 1. If the pyhin proteins inhibit transcription by sequestering SP1, he authors need to provide some explaination for the fact that the CMV promoter, which is activated by Sp1, appears resistant to the pyhin proteins (Figure 1D).

2. The RNAi knockdown experiment shown in Figure 3/Figure S1 is not compelling. The knockdowns are extremely variable, often not specific for the targeted protein and some knockdowns appear to affect cell viability (total cell protein). Some knockdowns even give the appearance of upregulating other Pyhin proteins, in an inconsistent manner. Overall this is a very unconvincing dataset. Since this experiment is the only one in the paper that does not involve overexpressed Pyhin proteins, this reviewer is very doubtful that pyhin proteins restrict HIV-1 infection when present at endogenous levels

3. The magnitude of the antiviral effect displayed by these proteins is modest (3 to 5-fold inhibition in most experiments), even when grossly overexpressed by transient transfection in 293T cells. Stable expression of these proteins in a T-cell line should be able to prevent HIV-1 replication in a viral growth assay, if the authors model is correct.

4. Figure 6 shows great variation in the sensitivity to pyhin proteins among HIV-1 strains which is unexplained. This result suggests that is it trivially easy for HIV-1 to escape the antiviral effects of pyhin proteins, which isn’t the expected property of a genuine restriction mechanism

5. The coprecipitation assays in Figure 6D and 7B showing that pyhin proteins bind to Sp1 are unconvincing – Controls are inadequate, there is a significant signal in the control lanes, and both active and inactive pyhin proteins apparently coprecipitate Sp1.

6. The experiments in Figure 6 are extremely superficial. Based on their varying susceptibility to inhibition by overexpressed Pyhin proteins it is claimed that the other viruses used in these experiments may be inhibited by different mechanisms. If so, then these experiments are irrelevant to the current study

Reviewer #3: 1. The conclusions regarding the mechanism of HIV inhibition (interference with Sp1-dependent gene transcription; competition of PYDs with DNA for SP1 binding) are entirely based on overexpression in 293T cells. The manuscript could be strengthened by evidence that this also occurs in a cellular context that is more relevant for HIV-1. 293T cells are not physiologically relevant target cells, and typically express unusually high protein levels upon transfection.

2. The biological relevance of the study hinges on the primary macrophage experiments (Fig. 3), for which only a single control siRNA was used. In the experience of this reviewer, certain “control siRNAs” can have significant effects on infectious HIV-1 yields, and the authors should exclude this possibility.

3. Fig. 5C: Why did the “IFI16+AIM2 linker” construct increase HIV-1 expression in a dose-dependent manner? This enhancement was almost as pronounced as the inhibition by native IFI16. Clearly, the AIM2 linker did not just disrupt the function of IFI16, as the authors state (line 192).

**Part III – Minor Issues: Editorial and Data Presentation Modifications**

Reviewer #1: 3. line 89, add “when over-expressed” in 293T cells, for clarity

4. Figures 1D, 5C, use symbols that are more different for clarity

5. Figure 5F, can we see some IF of the specks? Do they look like we expect?

6. In Figure 2 the authors say they made a standard curve with an IFN titration. Can we see the data plotted as IFN units which would be more meaningful. If this isn’t possible, that’s fine, but if they’ve gone to the trouble to drive the IFN values, I’d show them.

7. Do the authors think that AIM2 antiviral activity is through binding and titrating Sp1? Is this expected for a protein that doesn’t normally go into the nucleus? Can they clarify this point?

8 line 266 the authors are not investigating the effect of HBV causing 250 million infections, reword for clarity.

Reviewer #2: (No Response)

Reviewer #3: 1. Page 8, line 157; Fig. S1B: The statement that “the levels of MNDA expression were significantly reduced in virally infected cultures” is difficult to reconcile with the data shown in Fig. S1B

2. Fig. 3C: The Y axis seems to be mislabeled (% of ctrl siRNA). According to the text, the figure shows the effects of HIV-1 infection on protein expression levels.

3. Page 10, top: a subheading should be used for the section on inflammasome assembly

4. Fig. 6F: The IFI16-AIM2 linker construct should also be included here.

5. Page 12 and 13: Fig. 7A-G are all mislabeled (should be Fig. 8A-G).

PLOS authors have the option to publish the peer review history of their article (what does this mean?). If published, this will include your full peer review and any attached files.

Reviewer #1: No

Reviewer #2: No

Reviewer #3: No
---

## [Editor Report · Decision Letter 1]

26 Jun 2020

Dear Dr. Kirchhoff,

We are pleased to inform you that your manuscript 'Nuclear PYHIN proteins target the host transcription factor Sp1 thereby restricting HIV-1 in human macrophages and CD4+ T cells' has been provisionally accepted for publication in PLOS Pathogens.

Best regards,

Bryan R. Cullen

Associate Editor

PLOS Pathogens

Thomas Hope

Section Editor

PLOS Pathogens

Kasturi Haldar

Editor-in-Chief

PLOS Pathogens

orcid.org/0000-0001-5065-158X

Michael Malim

Editor-in-Chief

PLOS Pathogens

orcid.org/0000-0002-7699-2064
---

## [Editor Report · Acceptance letter]

30 Jul 2020

Dear Dr. Kirchhoff,

We are delighted to inform you that your manuscript, "Nuclear PYHIN proteins target the host transcription factor Sp1 thereby restricting HIV-1 in human macrophages and CD4+ T cells," has been formally accepted for publication in PLOS Pathogens.

Best regards,

Kasturi Haldar

Editor-in-Chief

PLOS Pathogens

orcid.org/0000-0001-5065-158X

Michael Malim

Editor-in-Chief

PLOS Pathogens

orcid.org/0000-0002-7699-2064